# Quantitative proteomic landscape of metaplastic breast carcinoma pathological subtypes and their relationship to triple-negative tumors

Sabra I. Djomehri [1,2,3], Maria E. Gonzalez[1,3], Felipe da Veiga Leprevost[1], Shilpa R. Tekula [1,3], Hui-Yin Chang[1], Marissa J. White[4], Ashley Cimino-Mathews[4], Boris Burman[1,3], Venkatesha Basrur [1], Pedram Argani[4], Alexey I. Nesvizhskii [1,5,6 ✉] & Celina G. Kleer[1,3,6 ✉]

Metaplastic breast carcinoma (MBC) is a highly aggressive form of triple-negative cancer (TNBC), defined by the presence of metaplastic components of spindle, squamous, or sarcomatoid histology. The protein profiles underpinning the pathological subtypes and metastatic behavior of MBC are unknown. Using multiplex quantitative tandem mass tag-based proteomics we quantify 5798 proteins in MBC, TNBC, and normal breast from 27 patients. Comparing MBC and TNBC protein profiles we show MBC-specific increases related to epithelial-to-mesenchymal transition and extracellular matrix, and reduced metabolic pathways. MBC subtypes exhibit distinct upregulated profiles, including translation and ribosomal events in spindle, inflammation- and apical junction-related proteins in squamous, and extracellular matrix proteins in sarcomatoid subtypes. Comparison of the proteomes of human spindle MBC with mouse spindle (CCN6 knockout) MBC tumors reveals a shared spindle-specific signature of 17 upregulated proteins involved in translation and 19 downregulated proteins with roles in cell metabolism. These data identify potential subtype specific MBC biomarkers and therapeutic targets.

[1] Department of Pathology, University of Michigan Medical School, Ann Arbor, MI 48109, USA. [2] Molecular Cellular Pathology Training Program, University of Michigan, Ann Arbor, MI 48109, USA. [3] Rogel Cancer Center, University of Michigan, Ann Arbor, MI 48109, USA. [4] Department of Pathology, Johns Hopkins University, Baltimore, MD 21287, USA. [5] Department of Computational Medicine and Bioinformatics, University of Michigan, Ann Arbor, MI 48109, USA. [6] These authors contributed equally: Alexey I. Nesvizhskii, Celina G. Kleer ✉email: nesvi@med.umich.edu; kleer@umich.edu

Triple-negative breast cancer (TNBC) comprise a heterogeneous group of tumors with varying histologies and prognosis[1]. The most lethal subtype of TNBC is termed metaplastic breast carcinomas (MBC), a unique and heterogeneous group that account for 0.2–5% of all breast cancers[2,3]. MBCs are characterized by the presence of a glandular epithelial component and a hallmark non-glandular metaplastic component that may consist of cells with spindle, squamous, and/or sarcomatoid (i.e., chondroid or osseous) features[4,5]. Clinically, MBCs are more metastatic and chemoresistant than non-metaplastic TNBC, with the spindle subtype reported to have the worst prognosis[6]. This significant clinical challenge highlights the need to distinguish MBC tumors for diagnostic and precision treatment purposes.

The molecular alterations that distinguish MBC from TNBC, and the protein profiles that determine MBC histological subtypes are poorly understood[7–9]. To date, few studies have investigated the genomic and transcriptomic features of MBC[10–12]. Genetically, MBCs have a high level of genomic instability, display a complex copy number variation pattern, and tend to harbor significantly more mutations in *PIK3CA*, *WNT*, and *TP53* compared with TNBCs, as well as presenting loss of *CDKN2A* and overexpression and amplification of *EGFR*[13–15]. Studies have been unable to show a genetic basis among MBC histologic subtypes, but have recently demonstrated distinct profiles at the transcriptomic level[10]. However, little is known about the mechanisms of MBC metastasis, and there are no reliable biomarkers or targets of therapy, underscoring the need to identify protein profiles specific to MBC and subtypes.

Our lab has generated MMTV-cre;Ccn6^fl/fl knockout mice which form mammary tumors that recapitulate high-grade human MBC with predominant spindle components both morphologically and transcriptionally[16]. Using this model, we have identified an 87-gene signature common between mouse and human MBCs including the transcription and translation regulatory proteins HMGA2 and IGF2BP2[17]. In addition, histological and immunophenotypical evidence suggests that MBCs are enriched in epithelial-to-mesenchymal (EMT) and different cellular compartment express stemness markers[18–20]. However, the protein profiles of human and mouse MBC are unknown.

Few proteomics studies to date explore invasive breast carcinoma signatures, with very limited information on MBC and normal tissue counterparts[21–23]. Here, we use human tissue samples of MBC, TNBC, and normal breast and a quantitative custom-built platform to test the hypothesis that the histological subtypes of MBC may have distinct protein profiles that may result in their pathological phenotypic diversity and aggressive clinical behavior. We employ tandem mass tag (TMT) based proteomics technology and Philosopher/TMT-Integrator, a computational pipeline that our group has recently developed and optimized for the analysis of large patient cohorts, to process the acquired mass spectrometry data. By unraveling the protein signatures within MBC and their relationship to TNBC and normal breast tissue, our study advances the understanding of the biology of MBC and provides potential diagnostic and prognostic markers, as well as testable targets of therapy specific to MBC pathological subtypes.

## Results

**Human samples and clinical data**. To elucidate the proteomic profile of MBC and understand the differences in protein expression with TNBC and normal breast tissues, we assembled a clinical cohort of 15 frozen MBC which were classified clinically according to their predominant metaplastic component into the following subtypes: spindle (*n* = 6), squamous (*n* = 4), and sarcomatoid (*n* = 5). In addition, we included 6 non-metaplastic TNBCs and 6 normal adjacent breast tissues (Table 1, Fig. 1a). All patients were women with a mean age of 55 years old (range 33–89 years old). The majority (14 of 15, 93.33%) of MBC and all TNBC were of histological grade 3 (of 3), all were negative for estrogen and progesterone receptor, and for HER2/neu overexpression. Of the 15 MBC, 11 (78.6%) were stage I/II and 3 (21.4%) stage III/IV at the time of diagnosis. Of the 6 TNBC, 5 (83.3%) were stage I/II and 1 (16.6%) stage III at diagnosis. At follow-up, 4 of 14 (28.6%) MBC, and 1 of 6 (16.66%) TNBC, developed distant metastasis to the lungs, liver, skin, and bone.

**The proteome of human metaplastic breast carcinoma**. We leveraged the increased throughput of multiplexed TMT 10-plex proteomics and our automated and robust computational data analysis pipeline to generate a quantitative proteome profile of 27 human tissue samples (Fig. 1b). We arranged the samples into three experimental groups appropriate for the 10-plex TMT isobaric labeling strategy (*n* = 10 samples per experiment; 9 tissue samples and 1 reference sample consisting of a pool of all 27 tissues) (Supplementary Table 1). The three proteomic TMT 10-plex experiments identified 82,251, 84,667, and 84,386 peptides, respectively, to a depth of 5798 unique proteins across all samples (1% protein and 1% PSM false discovery rate). We used the MSFragger and Philosopher tools (v20181128, github.com/Nesvi-lab/philosopher) for peptide identification, protein inference, FDR filtering, and extraction of quantification information form raw data, and TMT-Integrator for additional quality assessment and filtering, PSM selection, outlier removal, peptide-to-protein quantification roll-up, and normalization (Fig. 1b). We performed a gold standard data imputation method using the multivariate imputation by chained equations (mice)[24] package in R, to preserve statistical power and sample size by producing unbiased estimates of the missing values. Next, we performed standard batch correction in R, and the resulting expression matrix was used for all downstream analyses (Supplementary Fig. 1a–b).

Among the 5798 proteins found in the MBC proteome, 5635 unique proteins passed through quality filters in the TMT-Integrator algorithm, consistent with each sample. The distribution of all patient samples and principal component analysis (PCA) shows a clear distinction between normal breast and tumor proteomes (Fig. 2a). Unsupervised *k*-means clustering methods between MBC and TNBC proteomes and hierarchical clustering demonstrates modest distinction of MBC subtypes and TNBC, the latter overlapping most with spindle and squamous MBC (Fig. 2b, Supplementary Fig. 2). In addition, 1 of the 15 MBC samples was excluded from downstream analyses since it was confirmed by histology and proteomics that the piece cut for analysis contained only normal tissue.

General features of the MBC proteome relative to normal tissues demonstrate a global downregulation program involving major tumor suppressors, extracellular matrix activities, and wound healing responses according to the clustering analysis (Fig. 2b), while enrichment analysis reveals the top GO molecular function for upregulated MBC proteins is procollagen-proline dioxygenase activity, and aldehyde dehydrogenase activity for downregulated proteins (Supplementary Fig. 3a–c, Supplementary Table 2). The MBC proteome establishes with functional relevance that MBC tumors and normal tissues have distinct protein profiles and exhibit deregulation of tumorigenic pathways.

**The MBC proteome relative to TNBC and within MBC subtypes**. Next, we sought to test the hypothesis that MBC histopathological subtypes are associated with specific proteomic signatures. We grouped patient samples according to histological

**Table 1 Clinical and histopathological features of the tumors in our patient cohort.**

| Case # | Age | Tumor size (cm) | Diagnosis | Histologic subtype | Tumor grade | Stage at diagnosis | Distant metastasis | Time to metastasize (months) | Status at follow-up |
|---|---|---|---|---|---|---|---|---|---|
| 1 | 43 | 4 | MBC | Chondroid | 3 | T2N0 | No | | NED |
| 2 | 89 | 2.8 | MBC | Chondroid | 3 | T2Nx | Liver, lung | 12 | DOD |
| 3 | 46 | 1.6 | MBC | Chondroid | 3 | T1cN0 | No | | NED |
| 4 | 68 | 3 | MBC | Chondroid, osseous | 3 | T2N0 | No | | NED |
| 5 | 39 | 1.8 | MBC | Chondroid | 3 | T3 | No | | NED |
| 6 | 50 | 29 | MBC | Spindle, osseous | 3 | T4N0 | No | | NED |
| 7 | 52 | 2.7 | MBC | Spindle | 3 | T2Nx | Skin, abdomen | | AWD |
| 8 | 64 | 2 | MBC | Spindle | 3 | T1cNx | No | | NED |
| 9 | 68 | 5.5 | MBC | Spindle | 3 | T3N0 | No | | DWD |
| 10 | 38 | 3 | MBC | Spindle | 3 | T2N2 | No | At diagnosis | AWD |
| 11 | 47 | 4.2 | MBC | Spindle (partial squamous) | 2 | T2N0 | No | | NED |
| 12 | 60 | 6 | MBC | Squamous (partial spindle) | 3 | T3N1 | Liver, lung, bone | At diagnosis | DOD |
| 13 | 55 | 4 | MBC | Squamous (partial spindle) | 3 | T2N0 | No | | NED |
| 14 | 53 | 4.5 | MBC | Squamous | 3 | T2Nx | Lung | 6 | AWD |
| 15 | 81 | 10.1 | MBC | Squamous | 3 | T3N0 | No | | NED |
| 16 | 50 | 2.6 | TNBC | IDC | 3 | T2N1mi | No | At diagnosis | NED |
| 17 | 33 | 2.1 | TNBC | IDC | 3 | T2N0 | No | | NED |
| 18 | 37 | 2 | TNBC | IDC (apocrine) | 3 | T1cN0 | No | | NED |
| 19 | 57 | 5.9 | TNBC | IDC (neuro-endocrine features) | 3 | T3N1a | Liver | 15 | AWD |
| 20 | 53 | 2.9 | TNBC | IDC | 3 | T2N0 | No | | NED |
| 21 | 73 | 2.8 | TNBC | IDC (medullary features) | 3 | T2N0 | No | | NED |
| 22–27 | 33–73 | – | Normal | | | | | | |

IDC invasive ductal carcinoma, NED no evidence of disease, AWD alive with disease, DOD in hospice or dead of disease, DWD dead with disease.

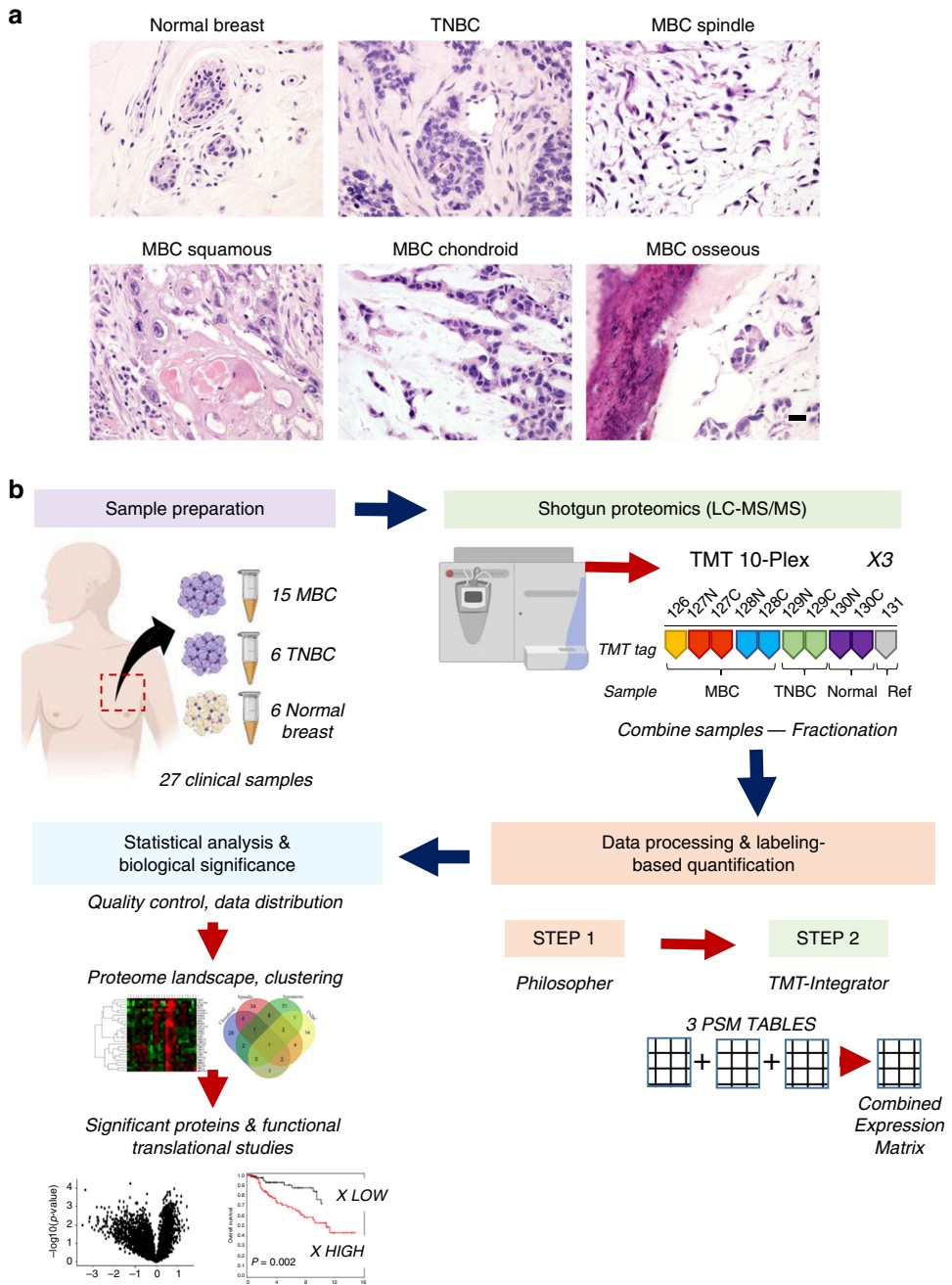

**Fig. 1 Human clinical samples and quantitative proteomics workflow. a** Representative images of hematoxylin and eosin stained tissues from our patient cohort, including 15 metaplastic carcinomas (MBC, 6 spindle, 4 squamous, and 4 sarcomatoid), 6 triple-negative (TNBC), and 6 normal breast. Scale bar = 50 μm. **b** Workflow of quantitative mass spectrometry profiling (cartoons created with BioRender.com). For data acquisition we assembled the 27 samples into 3 experimental groups for 10-plex LC-MS/MS tandem mass tag (TMT) isobaric labeling. For data processing and quantification, we used two computational pipelines, Philosopher/TMT-Integrator, and generated a combined protein expression matrix for the 3 experiments used for downstream analyses, including hierarchical clustering, differential expression tests, statistical analysis, and biological inference.

MBC subtypes, where samples with mixed features were grouped in the subtype belonging to their dominant histology (i.e., squamous with partial spindle features was grouped to squamous subgroup), and performed differential expression analysis of MBC relative to TNBC, and across MBC subtypes (Fig. 3). Compared with TNBC, MBC shows deregulation of the immune system (humoral immune responses; $p < 1e-04$) and extracellular structure organization. Our analyses also reveal distinct functional processes among MBC subtypes, such as enriched keratinization (epidermal and keratinocyte differentiation) in squamous,

regulation of proteolysis and protein activation cascade in spindle, and leukocyte activation and exocytosis in sarcomatoid (Fig. 3). These results uncover common as well as distinct cellular differentiation profiles within the MBC proteome.

To gain a better understanding of the functional differences within MBC pathological subtypes and elucidate potentially unique protein signatures, we applied gene set enrichment (GSEA) analysis from the molecular signature database (MSigDB) using hallmark (Fig. 4), canonical pathways (Supplementary Fig. 4, Supplementary Data 1–4) and GO gene sets (Supplementary Fig. 5,

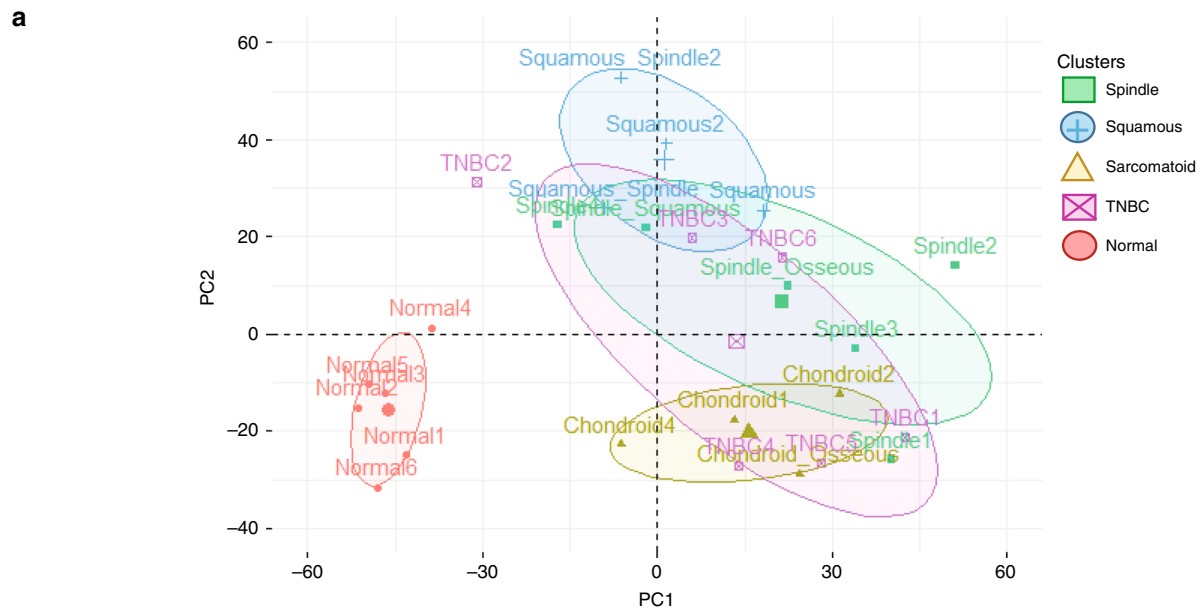

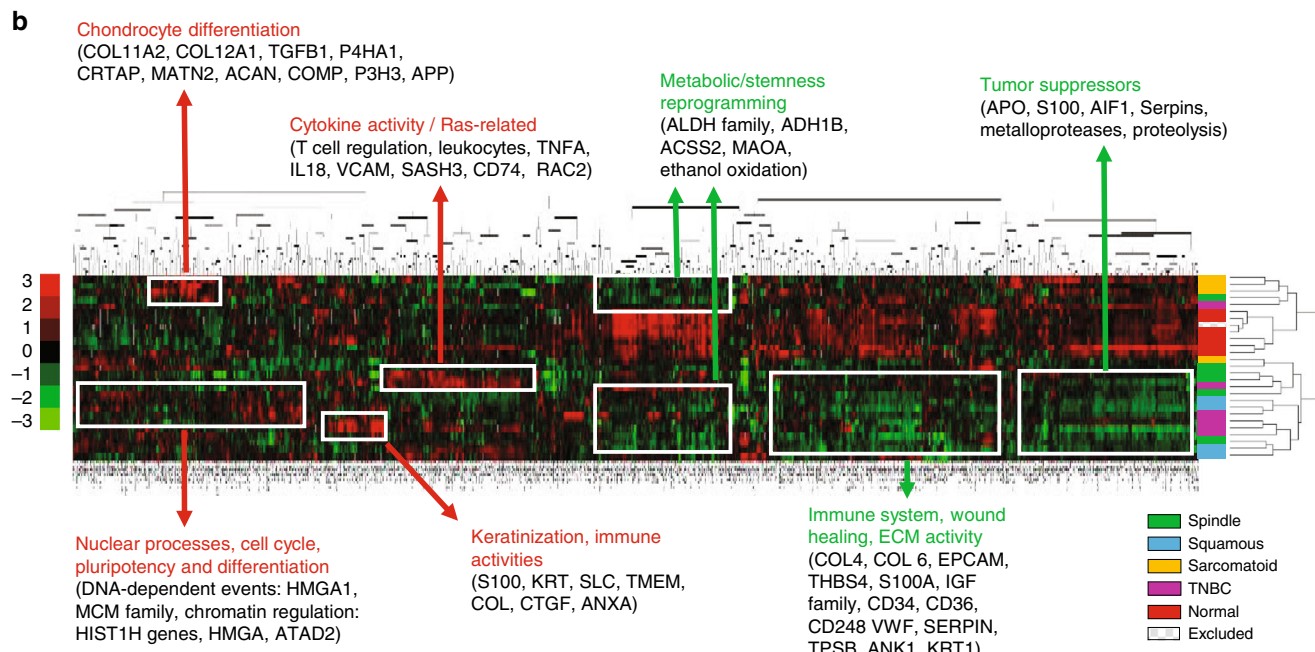

**Fig. 2 Quantitative proteomics of MBC, TNBC, and normal breast tissues. a** Principal component analysis (PCA) plot shows unsupervised clustering among the 27 samples, demonstrating a clear distinction between normal and all tumors (MBC subtypes and TNBC) and between MBC squamous and sarcomatoid, while there is an overlap between MBC subtypes and TNBC. **b** Heat map of the 5670 proteins that passed through quality filters in TMT-Integrator across all patients. Significantly enriched downregulated (green) and upregulated (red) GO biological processes ($p < 0.05$). Dendogram from hierarchical clustering analysis in Cluster 3.0 using median centering, uncentered correlation, and complete linkage, and visualized in Java TreeView 1.1.6r4. Scale bar shows expression level (color-coded as red for upregulated, green for downregulated, and black for unchanged.

Supplementary Data 1–4). Compared with TNBC, the top upregulated hallmark across MBCs is EMT while oxidative phosphorylation (OXPHOS) is the top downregulated hallmark pathway (Fig. 4).

GSEA hallmark analyses reveal distinct up- and downregulated protein profiles among MBC subtypes (Fig. 4), and GSEA enrichment plots delineate specific differences in top pathways (Fig. 5). Comparison within pathological subtypes show that spindle MBC has high MYC and E2F targets, and ribosome pathway proteins (KEGG pathways; Supplementary Fig. 4), squamous MBC has high interferon gamma (and broad inflammatory responses), TP53 and PI3K signaling, apical junction, and

low OXPHOS, MYC, and E2F targets, while sarcomatoid MBC has high EMT and OXPHOS, and low interferon gamma, MTORC1, and PI3K signaling (Figs. 4, 5). Visualization of top enriched up- and downregulated terms and their associated proteins are shown by protein networks (Supplementary Fig. 6) which can be investigated as potential therapeutic candidates. These up- and downregulated protein signatures were further validated by additional differential expression analyses of each MBC relative to TNBC (Supplementary Fig. 7). Together, these analyses pinpoint specific pathways which may operate in MBC compared with TNBC, and highlight MBC subtype-specific pathways for further functional investigations.

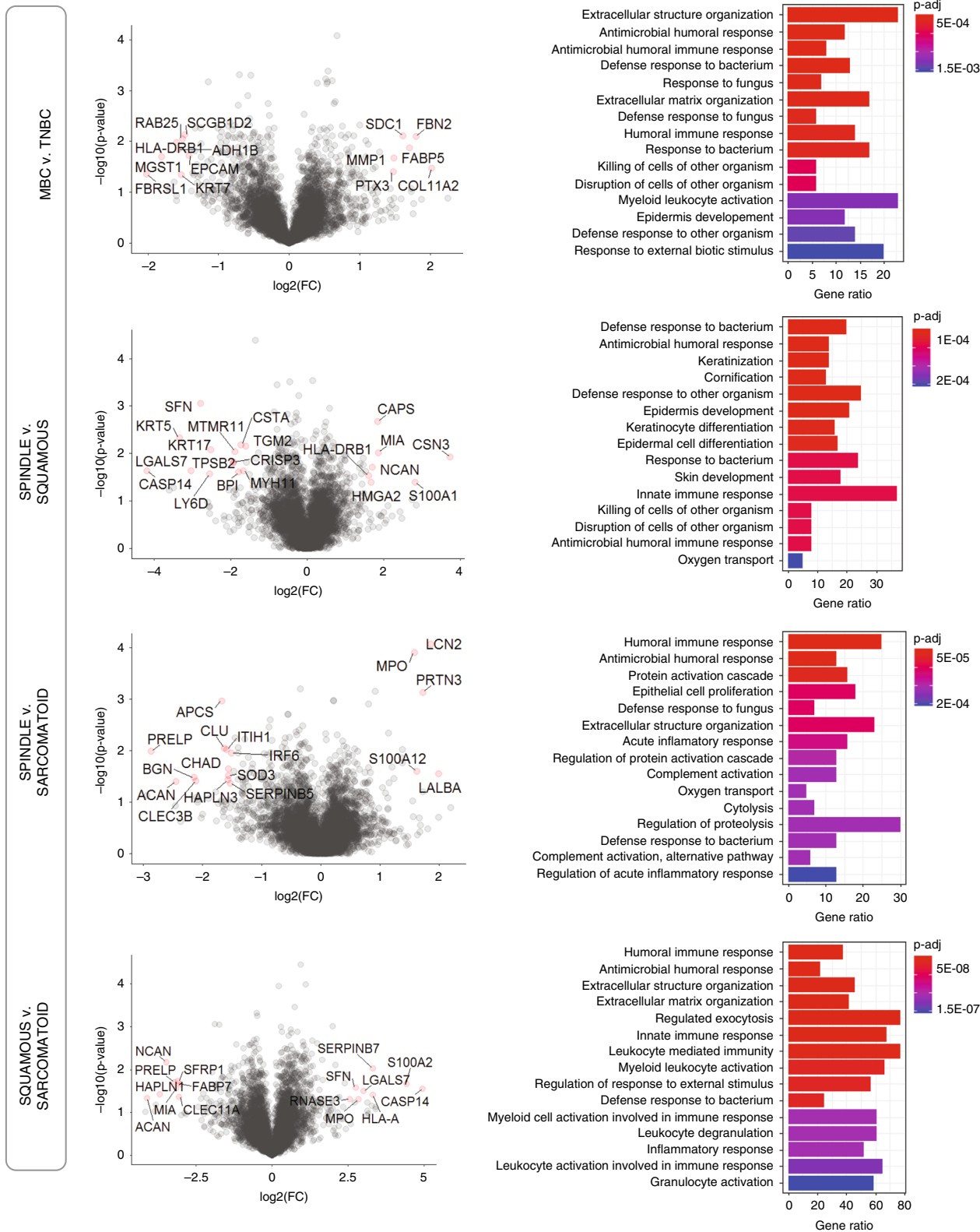

**Fig. 3 Differential expression analysis of histological MBC subtypes.** Left: Volcano plots comparing MBC with TNBC and within MBC subtypes, as indicated. Significantly differentially expressed proteins are highlighted in red. log2-Fold change versus −log10($p$-value), where cutoff values on plots show FC >1 or FC < −1 (vertical lines) and $p$ < 0.01 (horizontal line), $n$ = 20 patients. Right: Enrichment analysis using gene ontology (GO) annotations showing the top GO terms based on biological process, molecular function, or cellular compartment. Significant proteins were considered using $p$-value < 0.05, $q$-value < 0.1, fold change (log2-FC) >1, Benjamini–Hochberg correction (BH), gene set size 5–500, and the total protein list (5670 proteins) as the background set. Barplot shows significant terms by the gradient legend as $p$-adjust < 0.001, the $x$-axis is gene ratio and the number of proteins belonging to given enriched terms.

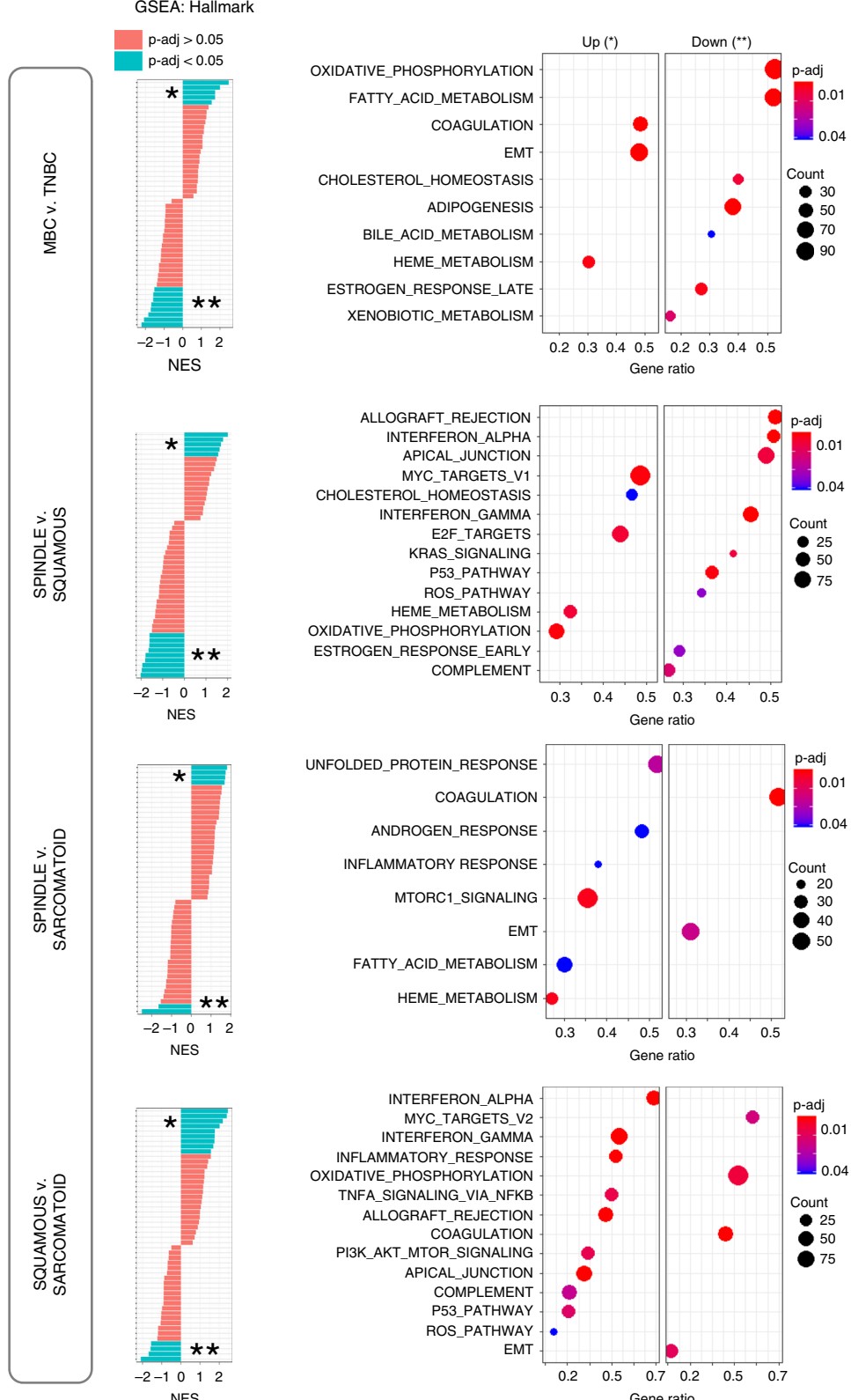

**Fig. 4 GSEA reveals hallmark pathways within MBC and relative to TNBC.** GSEA analyses show differentially expressed protein pathways in MBC compared with TNBC, and within MBC subtypes. Left: normalized enrichment scores (NES) versus the total list of GSEA hallmark upregulated (*) and downregulated (**) categories, with significantly enriched terms (p-adjust < 0.05) among non-enriched (p-adjust > 0.05). Right: only the top hallmark up- and downregulated pathways labeled as UP (*) and DOWN (**), highlighting unique significantly expressed categories, where the x-axis is gene ratio. The GSEA analysis was performed using the clusterProfiler and fgsea package in R for the hallmark collection (H) (Broad Institute), with n = 1000 permutations, where p-adjust < 0.05 and FDR < 0.05 were considered significant.

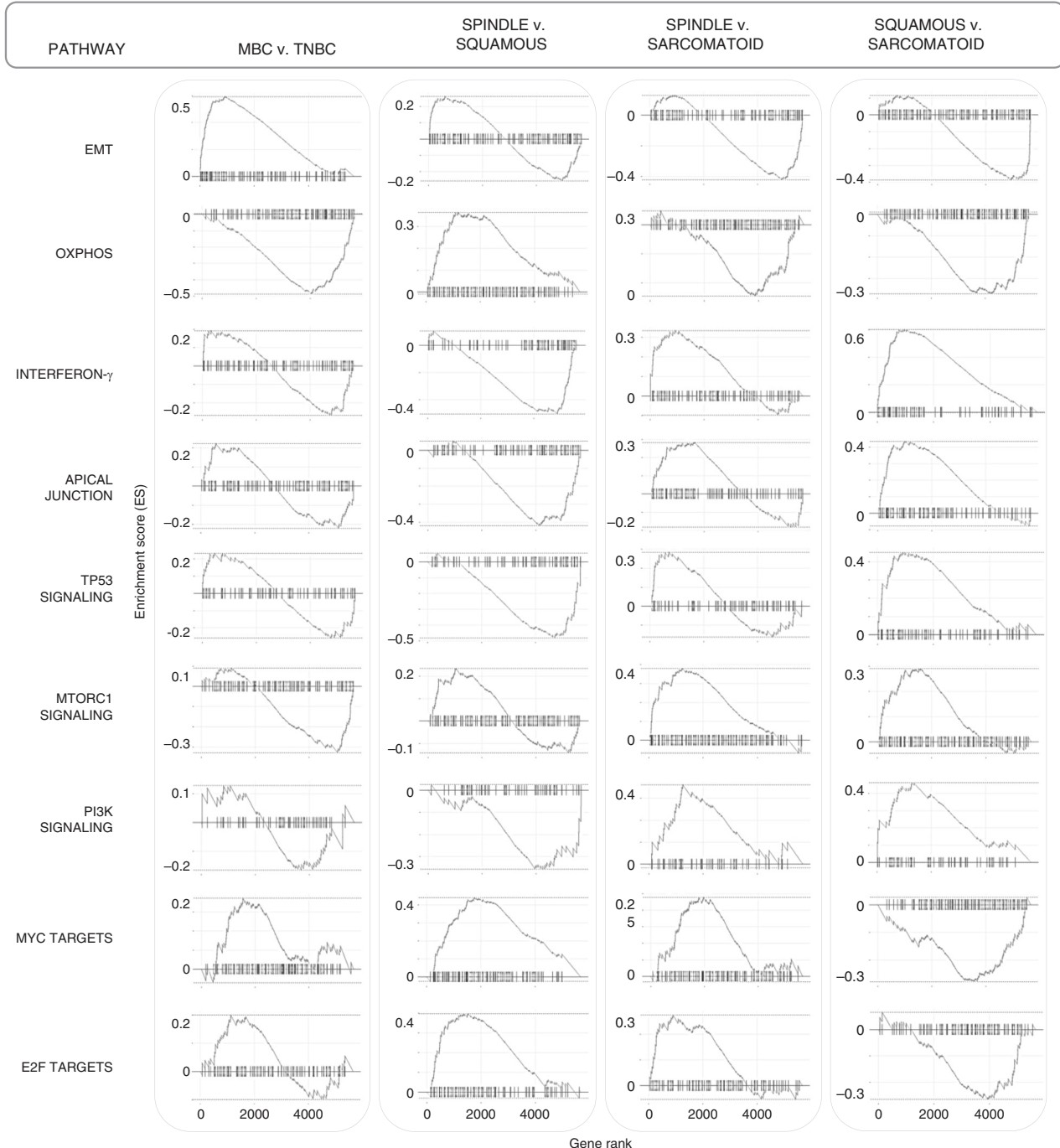

**Fig. 5 Top enriched pathway profiles distinguish MBC subtypes and TNBC.** GSEA enrichment plots of the highest up- or downregulated pathways by carcinoma type, including epithelial-mesenchymal transition (EMT), oxidative phosphorylation, interferon-γ, apical junction, TP53, MTORC1, and PI3K signaling, and MYC and E2F target pathways. The GSEA analysis was performed using the fgsea package in R for the hallmark collection (H) (Broad Institute), with $n = 1000$ permutations, where $p$-adjust $< 0.05$ and FDR $< 0.05$ were considered significant.

**MBC subtype-specific mutational signatures by WES analysis.** Based on our discovery of subtype-specific hallmark protein pathways in MBC, we set out to elucidate the hypothesis that each pathway may exhibit distinct DNA mutational profiles. Out of the 15 MBC tumors, 10 pairs of tumors with their normal tissue counterparts passed initial quality control for whole-exome sequencing (WES) (Fig. 6). Most of the variants observed were intronic (Fig. 6a), and when filtering only missense and loss-of-function or LoF (Fig. 6b), we found that MBCs share frequent somatic mutations in *TP53* (70%), *MUC17* (60%), *PLEC* (30%), *CRYBG2* (40%), and *ZNF681* (30%) (Fig. 6c). Spindle and

squamous tumors share *AHNAK* mutations (*AHNAK*, *AHNAK2* (80%) in spindle; *AHNAK* (33%), *AHNAK* (67%) in squamous) and *PI3K* family mutations (*PIK3C2A* (20%) in spindle; *PIK3CA* (33%) in squamous). Squamous MBC also harbor mutations in *MTOR*, *NOTCH3*, and *PTEN* (33%). Sarcomatoid MBC show genetic alterations in cadherin, calcium ion and *WNT* signaling, with frequent mutations in the protocadherin gene cluster (*PCDH* family), and *CDH7*, *MAP3K2*, and *FAT1* (50%) (Fig. 6b, e). In addition, allele frequencies are similar for variants across all subtypes (Fig. 6d). Taken together, these data show that MBCs share mutations in five genes, with only *TP53* having been

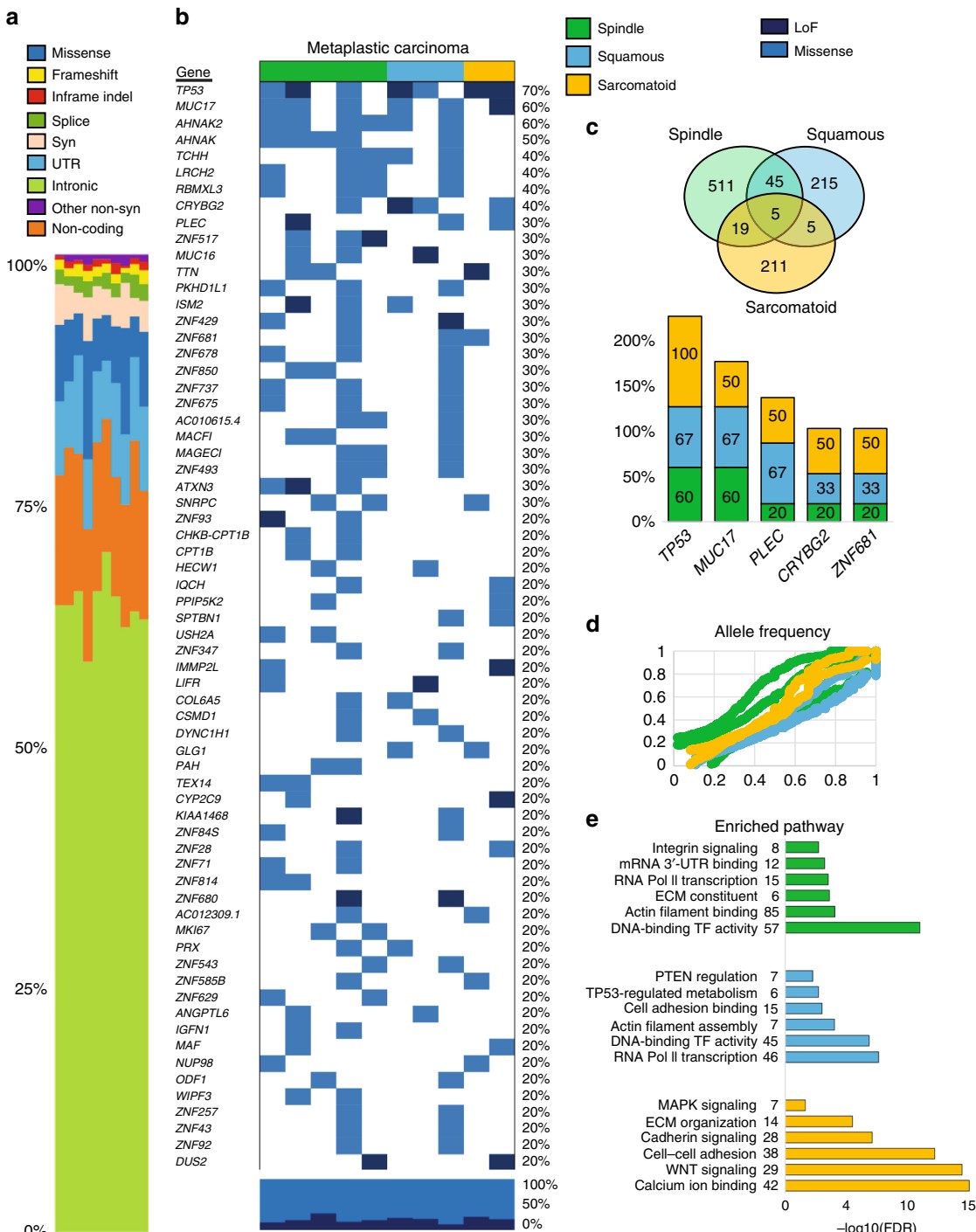

**Fig. 6 WES analysis shows repertoire of somatic mutations within MBC. a** The landscape of somatic mutations common to MBC and within subtype-specific histopathologies of 10 patients with MBC including spindle ($N = 5$), squamous ($N = 3$) and sarcomatoid ($N = 2$) and matching normal breast tissues. The type of mutation is color-coded as indicated in the legend. Pathogenic mutational variants in MBC were defined as those of high complexity (980 of 11,652 total) were filtered in this analysis. Syn: synonymous, INDEL: inframe insertions and deletions. **b** Heat map shows the top mutated genes with 20% of greater frequency of missense (blue) or loss-of-function mutations, LoF (dark blue). **c** Venn diagram highlights the number common and distinct mutated genes in MBC subtypes. Bars show the frequency of the five commonly mutated genes (*TP53*, *MUC17*, *PLEC*, *CRYBG2*, *ZNF681*) in each subtype. **d** Scatterplot is average allele frequency (AF) for each tumor sample, and colors represent metaplastic subtype. We excluded the variants with population allele frequency >5% based on 1000 Genome Project data. **e** Top enriched GO and pathways (KEGG, Panther, or Reactome) of mutated genes in MBC subtypes. Enrichment analysis was performed using gene lists extracted for each MBC according to the gene-level variant and effect summary analysis using GeneRollupv0.3.2. Variants that fell in low complexity genomic regions, genomic super DUPS, and repeat masker regions were excluded.

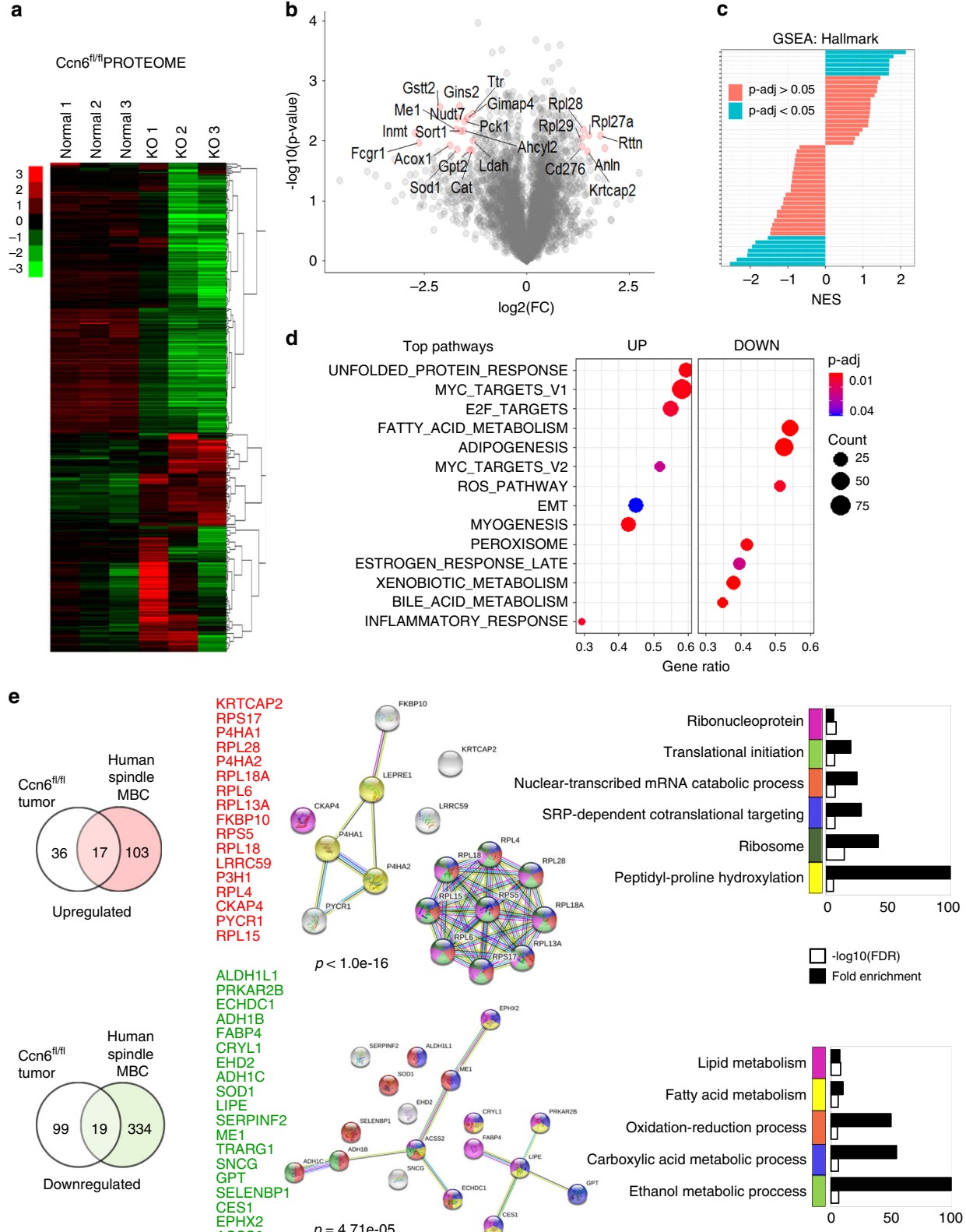

reported previously[25],[26]. These data also reveal that sarcomatoid MBC has a distinct mutational profile from spindle and squamous tumors, highlighting the importance of the proteomics landscape in distinguishing within pathological subtypes, which paves the way for mechanistic and functional studies elucidating the biology of these tumors.

**Comparison of human and mouse MBC proteomes.** To further refine the protein landscape of spindle MBC and determine common deregulated proteins between human and mouse tumors, we performed quantitative proteomics on MMTV-Cre;Ccn6$^{fl/fl}$ spindle MBC tumors followed by comparison with human MBC proteomes (Fig. 7a–c, Supplementary Fig. 8). Compared with normal mouse

**Fig. 7 Quantitative proteomics analysis of mouse MBC (MMTV-cre;Ccn6<sup>fl/fl</sup>). a** Heat map of the 4609 proteins that passed through quality filters in TMT-Integrator for all samples (3 MMTV-cre;Ccn6<sup>fl/fl</sup> mouse tumors (KO) and 3 normal mouse mammary glands). Scale bar shows expression level (red is upregulated and green is downregulated). **b** Volcano plot comparing Ccn6<sup>fl/fl</sup> tumors with normal mammary gland. Significantly differentially expressed proteins are highlighted in red. $p < 0.05$ and absolute value FC > 1 were considered significant. **c** Gene set enrichment analysis (GSEA) showing significant differentially expressed protein pathways in Ccn6<sup>fl/fl</sup> spindle MBCs compared with normal mammary glands. Normalized enrichment scores (NES) versus the total list of GSEA hallmark categories of up and downregulated hallmark pathways. **d** Top hallmark pathways highlighting significantly expressed up- and downregulated protein pathways (marked UP and DOWN, respectively). We used the molecular Signatures Database (MSigDB v7.0), hallmark gene sets with clusterProfiler and fgsea packages in R, with the biomaRT package in R to convert mouse gene IDs to human homolog associated gene symbols. **e** Venn diagrams demonstrate the overlap between the proteome of mouse MBC and human spindle MBC tumors, relative to their normal tissue counterparts, identifying a 17-protein upregulated and 19-protein downregulated protein set. Protein–protein interaction networks of up- and downregulated signatures highlight potential markers of interest, along with functional enrichment analysis using STRING v11.0. Color scheme of network proteins are matched by the legend of GO enrichment terms in barplots.

mammary glands, MMTV-Cre;Ccn6<sup>fl/fl</sup> tumors show increased MYC, E2F, unfolded protein response and ribosomal pathway proteins, which are similar to human spindle MBC (Fig. 7d).

Based on shared pathology, metastatic ability, and hallmark pathways, we hypothesized that human and mouse spindle MBC tumors may have an overlapping protein signature. Enrichment analyses in GO annotation and STRING databases uncover a set of 36 proteins, 17 upregulated and 19 downregulated, that overlap between human and mouse spindle MBC proteome (Fig. 7e). The shared downregulated proteins have roles in metabolic processes including oxidation-reduction, carboxylic acid and ethanol metabolism (e.g., ALDH1L1, ADH1B, ADH1C, SOD1, LIPE, FABP4). Among the upregulated proteins are those involved in ribosomal function, translation, and RNA metabolism (e.g., RPL18A, RPL18, RPL6). In addition, we found a highly enriched protein–protein interaction network among the upregulated proteins ($p < 1e-16$). Together, these data delineate significantly deregulated pathways in spindle MBC in human and mouse tumors and nominate a subset of proteins that may be useful markers and targets of therapy.

## Discussion

We present a quantitative proteomic landscape of human MBC, a subtype of TNBC with a defining histology, frequent chemoresistance, and distant metastases[27]. Using tissue resources from patients and from a spindle MBC mouse model, robust proteomics and bioinformatics tools we demonstrate shared and subtype-specific altered proteomes present in spindle, squamous and sarcomatoid MBC, providing insights into the biology of this aggressive form of breast cancer and offering opportunities for precision medicine.

Despite the impact of protein expression on tumor phenotypes and clinical behavior, our knowledge on the protein landscapes of human breast cancer, and how differential profiles contribute to breast cancer phenotypes is very limited. Recent advances in proteomic technology and bioinformatics have enabled detailed characterization of breast cancer[22,28]. However, studies to date have focused on frequent subtypes of breast cancer, with few to no MBC cases and normal breast tissues.

Our initial unsupervised clustering among the samples correlated with the tumor histopathology demonstrating a clear distinction between normal and all tumors (MBC subtypes and TNBC) and between MBC squamous and sarcomatoid, while there was an overlap between MBC subtypes and TNBC, supporting the hypothesis that MBC and TNBC have both common and distinct proteomic profiles that have not been characterized so far. Upon in-depth cross-analyses of MBC to both TNBC and normal tissues, and between each MBC subtype using GO and MSigDB databases, common and subtype-specific differences within MBCs and in relationship to TNBC emerged. Compared

with TNBC, the proteome of MBC has a highly enriched EMT phenotype, with increased inflammatory responses (mainly in spindle and squamous), an active ECM, and reduced oxidative phosphorylation. These data are consistent with our previous studies showing that MBC express proteins involved in the EMT process, which may contribute to a more stem-like and aggressive phenotype than TNBC, and with transcriptome studies showing that MBCs cluster as basal-like and claudin-low[18,29]. Subtype-specific proteins and pathways emerged when we compared proteomic profiles of spindle, squamous and sarcomatoid tumors. A salient feature of spindle MBCs, which are highly proliferative and exhibit pathological evidence of EMT with elongated cancer cells, is upregulated expression of E2F and MYC pathway proteins, ribosomal proteins and transcriptional and translational processes. These findings are intriguing based on data showing that E2F and MYC are key drivers of cancer cell proliferation and EMT processes, where ribosome biogenesis and increased translation play an essential role[30,31]. These data also suggest that spindle MBC have a deregulated balance between translation and metabolic pathways, which needs to be further investigated.

Our data show that squamous MBC has upregulated inflammatory responses (e.g., IFN-γ, TNFα, and PI3K/MTOR), keratinization, and widespread cell adhesion marker expression (e.g., apical junction, adherens, CAMs, and cytoskeleton proteins), and decreased oxidative phosphorylation, MYC, and E2F pathways relative to other MBC subtypes. On the other hand, sarcomatoid MBC, which includes chondroid and osseous differentiation, exhibit a predominance in extracellular matrix signaling cascades and an amplified EMT program, increased oxidative phosphorylation, and decreased inflammatory responses compared with both spindle and squamous subtypes, likely owing to its differentiation along mesenchymal lineages[32]. Also, we observed that in general, there is no significant difference in TP53 and PI3K pathways between MBCs and TNBC. Collectively, our proteomic analysis suggests that while spindle, squamous and sarcomatoid MBCs and TNBC may share initial neoplastic events, each MBC subtype appears to have unique and active differentiation programs.

MBCs are chemoresistant and metastatic, but the underlying molecular determinants and driver pathways are unclear. Further, there are no effective treatments against these tumors[2,33]. The heterogeneity of MBC has been investigated at the genetic level[10,13,34], however, the relationship between genomic alterations and proteomic profiles is unknown. While MBCs were reported to harbor somatic mutations in *TP53*, *PI3K/MTOR*, and *WNT* signaling pathway genes[10], no subtype-specific mutational profiles in MBCs have been shown to date. Our whole-exome sequencing analyses of paired tissue samples of MBC and normal breast from the same patients identified somatic mutations common to all MBCs in five genes; *TP53* in 70%, *MUC17* in 60%, *CRYBG2* in 40%, *PLEC* in 30%, and *ZNF681* in 30%. Of these,

mutations in *MUC17*, *CRYBG2*, *PLEC*, and *ZNF681* are novel in MBC. We found that while spindle and squamous MBC exhibit overlapping mutational profiles of genes involved in transcription, RNA metabolic processes and actin filament binding, sarcomatoid tumors harbor distinct mutations, especially in *MAPK*, *WNT*, protocadherin cluster genes, calcium binding, and ECM organization. Taken together, we discovered distinct somatic mutational profiles in MBC tumors, highlighting that of sarcomatoid MBC compared with the overlapping landscape of spindle and squamous tumors. These data underscore the relevance of elucidating the proteomic landscape to nominate subtype-specific proteins and pathways, especially between tumors with spindle and squamous differentiation that may lead to effective treatment targets.

Quantitative proteomics of our mouse model of spindle MBC MMTV-Cre;Ccn6<sup>fl/fl</sup> tumors demonstrated a significant overlap with human spindle MBCs, with shared enrichment in E2F, MYC, EMT, and ribosomal proteins. Our analyses further pinpoint a set of 36 proteins commonly deregulated in mouse and human spindle MBCs which have not been previously considered in these tumors. The 19 downregulated proteins have functions in metabolic processes, while of the upregulated the majority are ribosomal proteins, and those involved in transcription and translation. Together these data validate the MMTV-Cre;Ccn6<sup>fl/fl</sup> model as a useful tool to investigate and test therapeutic targets for spindle MBC, and identify a set of significantly deregulated proteins and pathways as therapeutically operable targets. Our studies may pave the way toward mechanistic and functional investigations of these proteins in the biology of spindle MBC.

In addition to subtype-specific tumor signatures, this study uncovered a stromal cell expression pattern in MBCs, with the most prominent being downregulated expression of mesenchymal stem cell proteins (CD59, CD248), macrophages (CD209), immune cells (CD8A), hematopoietic stem/endothelial progenitors/macrophages (CD34, CD36, CDH5), a recently discovered mammary stem cell marker (CDH5)[35], and detected evidence of endothelial-to-mesenchymal transition (EndMT; CDH5<sup>lo</sup>IGFBP4<sup>lo</sup>CD34<sup>lo</sup>). The stromal profile is intriguing and has not been previously considered in MBC.

In summary, this patient cohort identifies a common proteomic landscape of MBC in relation to TNBC, and highlights the existence of specific protein profiles underlying the different histopathological subtypes of human MBC. We show that quantitative proteomics refines the mutational landscape of MBCs and allows for further distinction of the tumor subtypes. We provide evidence for a significant overlap between MMTV-Cre;Ccn6<sup>fl/fl</sup> spindle MBC mouse model and human disease, which together with a common histopathology and transcriptome, validate this mouse model to test new treatments and investigate the mechanisms leading to the development of spindle MBC. The subtype-specific proteomes of MBC tumors emphasize a unique opportunity to impact precision therapies to improve the survival of women with this aggressive form of breast cancer.

## Methods

**Tissue samples, histology and pathological evaluation.** This research complies with the ethical regulations for work with tissue samples and for animal studies. The work with tissue samples was approved under IRB protocols NA84719 and NA41867 from Johns Hopkins University and HUM00050330 from the University of Michigan. The animal work was approved by the University of Michigan UCUCA under protocol PRO00009007.

We employed frozen tissue samples from 14 women with MBC and adjacent normal tissues from the surgical pathology files at Johns Hopkins University, 6 TNBC and corresponding adjacent normal tissues from the surgical pathology files at the University of Michigan. Each specimen was collected 30 min after operation and immediately transferred to sterilized vials, snap frozen in liquid nitrogen and stored at -80 °C. Tumors were diagnosed according to the World Health Organization (WHO) classification into the following groups: spindle, squamous,

and sarcomatoid (chondroid and/or osseous) MBC[36]. Other pathological and clinical features such as estrogen and progesterone receptor analysis, Her2/Neu expression, Ki-67 proliferation index, tumor grade, tumor size, lymph node and distant metastasis were recorded. At the time of sample preparation, we cut a representative 0.5 cm piece of each of MBC, TNBC and normal sample, which was embedded in paraffin, sectioned to 5 μm and stained with hematoxylin and eosin stained (H&E) for diagnosis confirmation.

**Sample preparation.** Frozen specimens were kept on dry ice and cut to ~50 mg of tissue from each sample to be used for proteomic analyses. Specimens were diced and mechanically dissociated with a scalpel and placed in labeled 1.5 ml Eppendorf tubes containing three stainless steel microbeads. Tubes were then submerged in liquid nitrogen for 60 s and immediately homogenized using a mixer mill (Retsch MM400) for 2–3 cycles at maximum speed (30 Hz vibrational frequency) at 60 s per cycle. If any tissue had not been homogenized, another cycle was repeated. Samples were then placed on ice and 495 μl of RIPA buffer and 5 μl protease inhibitor were added to each tube, resuspended, and placed on a rocker for 30 min on ice at 4 °C. Beads were removed from tubes with a magnet and samples were ultrasonicated and centrifuged at 15,805 × *g* for 30 min at 4 °C. The supernatant was collected to retrieve a desired 2 mg/ml of protein, followed by standard protein quantification methods and storage at −80 °C.

**Protein extraction and TMT labeling procedure.** Tandem mass tag (TMT) labeling was performed for mass spectroscopy (MS) using three consecutive TMT-10-plex isobaric labeling kits (ThermoFisher, Cat #90111) according to the manufacturer's protocol. A master mix containing equal amount of protein from each of the 27 samples was generated. 60 μg of protein from each sample and the master mix were reduced with DTT (5 mM) at 45 °C for 1 h followed by alkylation with 2-chloroacetamide (15 mM) at room temperature (RT) for 30 min. Proteins were precipitated by adding six volumes of cold acetone and incubating overnight at −20 °C and pelleted by centrifuging at 8000 × *g* for 10 min at 4 °C. Supernatants were discarded and pellets resuspended in 100 μl of 100 mM TEAB, digested overnight at 37 °C by adding 1.1 μg of sequencing grade modified porcine trypsin (Promega, V5113). TMT reagents were reconstituted in 40 μl of anhydrous acetonitrile and digested peptides transferred to the TMT reagent vial, and incubated at RT for 1 h. TMT channels for each of sample are given in Table 1. The reaction was quenched by adding 8 μl of 5% hydroxylamine and incubating for 15 min. For each of the three TMT experiments, 9 samples and 1 master mix (Table 1) were combined, dried, followed by 2D separation, with the first dimension containing an aliquot from each sample mix (100 μg) underwent fractionation using high pH reverse phase fractionation kit (following the manufacturer's protocol, Pierce). Fractions were then dried and reconstituted in 10 μl of loading buffer, 0.1% formic acid and 2% acetonitrile.

**LC-MS/MS analysis.** For data acquisition, an Orbitrap Fusion (ThermoFisher) and RSLC Ultimate 3000 nano-UPLC (Dionex) was used to obtain raw data. To increase accuracy and confidence in protein abundance measurements, a multinotch-MS3 method was employed for MS data analysis. Two microliters from each fraction were resolved in 2D on a nanocapillary reverse phase column (Acclaim PepMap C18, 2 micron, 75 μm i.d. × 50 cm, ThermoFisher) using a 0.1% formic/acetonitrile gradient at 300 nl/m (2–22% acetonitrile in 150 m, 22–32% acetonitrile in 40 m, 20 min wash at 90% followed by 50 min reequilibration) and directly sprayed onto Orbitrap Fusion with EasySpray (ThermoFisher; Spray voltage (positive ion) = 1900 V, Spray voltage (negative ion) = 600 V, method duration = 180 min, ion source type = NSI). The mass spectrometer was set to collect the MS1 scan (Orbitrap; 120 K resolution; AGC target 2 × 10⁵; max IT 100 ms), and then data-dependent Top Speed (3 s) MS2 scans (collision induced dissociation; ion trap; NCD 35; AGC 5 × 10³; max IT 100 ms). For multinotch-MS3, the top 10 precursor ions from each MS2 scan were fragmented by HCD followed by Orbitrap analysis (NCE 55; 60 K resolution; AGC 5 × 10⁴; max IT 120 ms; 100-500 *m/z* scan range).

**Data analysis and protein quantification.** Raw MS data were converted using msconvert in Proteowizard software suite[37] to mzML format. MS/MS spectra were searched using the MSFragger (v20181128) database search tool[38] against UniProt human protein database (UP000005640, last modified: 13 December 2018; 73,928 proteins), appended with an equal number of decoy sequences and common contaminants. MS/MS spectra were searched using the following criteria: precursor-ion mass tolerance of 20 ppm, fragment mass tolerance of 0.6 Da (C12/C13 isotope errors (−1/0/1/2/3)), where cysteine carbamylation (+57.0215) and lysine TMT labeling (+229.1629) were specified as fixed modifications, and methionine oxidation (+15.9949), N-terminal protein acetylation (+42.0106), and TMT labeling of peptide N-terminus and serine residues were specified as variable modifications. The search was restricted to fully tryptic peptides, allowing up to two missed cleavage sites. MSFragger output files were processed using Philosopher toolkit (v20181119, github.com/Nesvilab/philosopher) as follows: The search results were first processed with PeptideProphet[39] (high-mass accuracy binning, semi-parametric mixture modeling options); ProteinProphet[40] was used to assemble peptides into proteins (protein inference) to create a combined file of

high-confidence proteins. Protein groups were filtered to 1% false discovery rate (FDR) using the target-decoy strategy and the best peptide approach[41,42], picked FDR adjustment[43]. The individual PSM lists for each TMT 10-plex we assembled, with peptides assigned either as a unique peptide to a particular protein group or assigned as a razor peptide[44] to a single protein group that had the most peptide evidence, and additionally filtered to 1% PSM-level FDR. For PSMs passing these filters, MS1 intensity of the corresponding precursor-ion was extracted using the Philosopher label-free quantification module based on the moFF method (10 ppm mass tolerance and 0.4 min retention time for extracted ion chromatogram peak tracing)[45]. For all PSMs corresponding to a TMT-labeled peptide, 10 TMT reporter ion intensities were extracted from the corresponding MS3 scans (using 0.002 Da window) and precursor-ion purity scores were calculated using the intensity of the sequenced precursor ion and that of other interfering ions observed in MS1 data (within 0.7 Da isolation window).

**Normalization and quality control.** Three PSM tables from each TMT-10-plex experiment generated as Philosopher output files were used as input to TMTIntegrator (v1.0.0; github.com/huiyinc/TMT-Integrator) for normalization and generation of integrative reports at the gene and protein level. For best quantitation quality, we only included PSMs (unique and razor peptides) that passed the following criteria: TMT label and quantification in the reference sample exist, minimum peptide probability 0.9, precursor ion purity ≥ 50%, minimum MS1 intensity 0.05%, summed MS2 intensity ≥ 5%. PSMs mapping to common contaminants were excluded, and for redundant PSMs, a single PSM of highest summed TMT intensity was kept. Then, intensities in each TMT channel were log2 transformed, and the reference channel intensity (pooled reference sample) was subtracted from that for the other nine channels (samples), thus converting the data into log2-based ratio to the reference scale. Here, the master mix TMT channel (131) was used as the reference channel. PSMs were grouped by the corresponding gene/protein, and outlier removal using the interquartile range (1.5 IQR) algorithm was applied.

The gene-level median was calculated from the remaining PSM ratios, and then normalized using the median absolute deviation (MAD), given the $p$ by $n$ table of ratios for entry $j$ in sample $i$, $R_{ij}$, the median ratio $M_i = \mathrm{median}(R_{ij}, j = 1,\ldots,p)$, and the global median across all $n$ samples, $M_0 = \mathrm{median}(M_i, i = 1,\ldots,n)$. The ratios in each sample were median centered, $R^C_{ij} = R_{ij} - M_i$. The median absolute deviation of centered values in each sample was calculated, $\mathrm{MAD}_i = \mathrm{median}(\mathrm{abs}(R^C_{ij}), j = 1\ldots p)$, along with the global absolute deviation, $\mathrm{MAD}_0 = \mathrm{median}(\mathrm{MAD}_i, i = 1,\ldots,n)$, and scaled to derive the final normalized ratios: $R^N_{ij} = (R^C_{ij}/\mathrm{MAD}_i) \times \mathrm{MAD}_0$. Finally, normalized ratios were converted back to the absolute intensity scale using the estimated intensity of each entry in the reference sample. The reference intensity was estimated using the weighted sum of MS1 intensities of the top 3 most intense peptide ions[46], and in computing $\mathrm{REF}_i = \mathrm{mean}(\mathrm{REF}_{ik}, k = 1,\ldots, q)$, missing intensity values were imputed with a global minimum intensity value. The final abundance (intensity) of entry $i$ in sample $j$ (log2 transformed) was computed as $A_{ij} = R^N_{ij} + \log_2(\mathrm{REF}_i)$. The tutorial describing all steps of the analysis, including specific input parameter files, command-line option, and all software tools necessary to replicate the results are available at the following website (github.com/Nesvilab/philosopher). Peptide assignments to MS/MS spectracan be visualized using PSM tables and the corresponding mzML files using freely available PDV spectrum viewer[47].

**Missing data imputation.** Upon inspection, experiments 1, 2, and 3 contained 18%, 14%, and 12% missing values, respectively (Supplementary Fig. 1). For missing data imputation, we used the multivariate imputation by chained equations algorithm using the statistical software 'mice' package (v3.4.0) in R is a gold standard method of choice to handle missing data and has been reported to produce lower estimate error rates compared with other methods (e.g., $K$ nearest neighbors or singular value decomposition). We used the following input parameters: $m = 5$ (number of imputed data sets), method='pmm' (predictive mean matching), and maxit = 50 (number of iterations). Rubin's Rules[24] were applied to pool estimates using logistic regression modeling. For statistical performance, we used significance testing of the categorical variables (i.e., patient expression values) to generate a covariance matrix containing pooled regression coefficient estimates, standard errors of parameter estimates, and $p$-values. We derived a pooled $p$-value < 1.6e−03 using the median $p$-value from the significance tests. The computational details are described in the original Rubin–Barnard approach[24].

**Statistical analysis and subtyping of MBC.** The batch corrected, imputed, combined data matrix was loaded and unsupervised PCA clustering was performed in R (v3.4.0) using standard $k$-means algorithms (e.g., Bayesian, Silhouette, Elbow) for finding optimal clusters, Cluster 3.0 was used for hierarchical clustering (median centering, uncentered correlation, and complete linkage), and visualization of results using Java TreeView (1.1.6r4). For differential expression analysis between tumors (MBC, TNBC) and normal samples, where data for each patient subgroup (e.g., spindle, triple-negative, normal) were pooled and results averaged, fold change was calculated by subtraction of averaged tumor subgroups to averaged control, and a Student's $t$-test was used to estimate $-\log_{10} p$-value. Proteins within a statistical region of FC > 1 and $p < 0.05$ were considered for analysis, including all

proteins with FC > 2 and $p < 0.01$. For enrichment analyses, we performed gene ontology (GO) over-representation tests (GO annotations: biological process, molecular function, cellular compartment, protein domain) in PANTHER (v14.1), and STRING (v11.0) database for confirming enrichment results with topological features from protein–protein interaction networks among subgroups. For patient stratification tests, one-way ANOVA statistical analysis was performed between multiple groups and $p < 0.05$ was considered significant.

**Gene set enrichment analysis (GSEA).** GSEA is an aggregate score and running-sum statistic approach that enables molecular signature based statistical significance testing that considers the entire gene set containing a ranked list of all expression values in a data set without requiring a cutoff of differentially expressed values for functional analysis[48,49]. We supplied a pre-ranked list of two classes, up- and downregulated fold change values for each subgroup to be analyzed (MBC vs. TNBC, and across MBC subgroups: Spindle vs. Squamous, Spindle vs. Sarcomatoid, and Squamous vs. Sarcomatoid). To understand functional enrichment profiles of the proteomics results, we used annotated gene collections downloaded from the Molecular Signatures Database (MSigDB v7.0 for H (hallmark gene sets), C2 (curated gene sets), and C5 (GO gene sets)[50]. We determined normalized enrichment scores (NES) with the total protein list as background, and the following parameters: $n = 1000$ permutations, where $p$-adjust < 0.05, and FDR < 0.05 were considered significant. The GSEA analysis was performed using the clusterProfiler and fgsea package in R and loading gene set collections from available gmt files from the BROAD Institute according to GSEA documentation[49,50].

**Whole-exome sequencing (WES).** DNA samples with matched tumor and healthy counterparts from 10 pairs of samples in our patient cohort was subjected to whole-exome sequencing using the NovaSeq 6000 Illumina system at the University of Michigan Advanced Genomics Core Facility in 150 bp paired-end format. Libraries were prepared used the NEBNext Ultra 2 FS DNA library prep kit for Illumina (NEB #E7805S; New England BioLabs) with 100 ng DNA input, 15 min fragment, 275–475 bp size selection and 6 PCR cycles. Samples were captured with the IDT xGen hybridization capture kit using 174 ng of each library pooled for capture and a final PCR of 7 cycles. Fastq generation was performed using Illumina's bcl2fastq software version 2.20.

**WES analysis and pathway enrichment.** The exome sequencing data was analyzed by the variant calling pipeline developed by the University of Michigan Bioinformatics Core. FastQC v0.11.7 was used to assess the quality of raw reads, which were trimmed to remove Illumina adapters and low quality ends using Trimmomatic v0.39, aligned to the hg38 reference genome using BWA v0.7.17, followed by removal of sequence duplicates, SAM tag fixing, local realignment around INDELs, base quality score recalibration and target coverage summarization using GATK v4.1.4.0 or v3.8 (for indel realignment only). Normal-Tumor paired alignment files were submitted to GATK's MuTect2, Varscan v2.4.4, and Strelka v2.9.10 for the detection and filtration of somatic variants. Only the somatic variants on the canonical chromosomes that passed each caller's quality filter were kept. For Mutect2, variants were first called from the normal samples and a panel of normals (PoN) was created and used for somatic variant calling. Cross-sample contamination and orientation bias were estimated and used for variant filtration. For VarScan, an alignment coverage file was first created using SAMtools v1.5 from the pair of normal and tumor samples together, and then used for somatic variant calling. The selected high-confidence somatic variants were filtered using VarScan's fpfilter after allelic read counts were calculated using bam-readcount v0.8. For Strelka, structural variant/INDEL candidates were first called using Manta, as recommended by Strelka manual as a best practice.

Candidate variant calls across all samples and patients were merged using Jacquard v1.1.2 and included all variant loci whose filter field passed in MuTect2 or Strelka or VarScan (VarScan calls were limited to high-confidence somatic variants confirmed in false-positive filter). Allele frequency (AF) parameter from Strelka was calculated by Jacquard based on the allelic read depth reported by Strelka since Strelka does not report AF directly. Jacquard inferred consensus genotype (GT), and calculated average alternate allele frequency (AF), from individual caller's calling results. Variants were annotated using VarSeq v1.4.3. Variants excluding intergenic variants and common SNPs (SNPs whose allele frequency is higher than 5% in 1000 Genomes Phase 3 data set) were integrated to create a gene-level variant and effect summary table using GeneRollup v0.3.2. Before filtering was applied, there were 11,652 total variants including those that fell in low complexity genomic regions, common SNPs, and duplicated regions. After filtering only out low complexity genomic regions, there were 980 total variants.

For enrichment analyses, gene lists of somatically mutated genes were generated for each MBC subtype and subjected to GO and pathway (KEGG, Panther, or Reactome) enrichment analysis using the WEB-based GEne SeT AnaLysis Toolkit[51]. We applied BH correction, at least three genes per pathway, $p$-adjusted < 0.05 as significantly enriched, and used the human genome as the reference set.

**MMTV-cre;Ccn6$^{fl/fl}$ mouse sample preparation and analysis.** Mouse mammary glands were harvested at necropsy following The University of Michigan approved protocol PRO00009007. Fresh mammary tumors and normal tissues were collected

and samples prepared for LC-MS/MS analysis using the same procedure as done for human samples, detailed above. One ~0.3 cm tumor section was embedded in paraffin, stained by hematoxylin and eosin and evaluated pathologically by the authors (C.G.K., a board certified pathologist) for tissue diagnosis. One proteomic TMT 10-plex experiment identified 85,683 peptides to a depth of 4609 unique proteins across all samples ($n = 10$ samples, where 6 were included for this analysis; 3 normal mouse tissues and 3 CCN6 knockout tumors). Pre-processing in MSFragger and Philosopher were performed the same as for human samples, but here we used the UniProt mouse protein database (UP000000589, last modified: 5 November 2019; 55,408 proteins). The TMT-Integrator analysis was performed the same as for human samples, however, a master mix sample was not required for a single TMT 10-plex experiment. Instead, we pooled the intensities from all samples (all log2 transformed TMT channels) and used this as the reference channel. In addition, data imputation was not necessary for a single TMT 10-plex experiment and we confirmed the average percentage of missing values was very low at <0.03%, hence, the missing values were simply removed from the data set. Hierarchical clustering, enrichment and GSEA analyses were performed as the same as indicated above. For overlapping the protein profiles of mouse and human spindle MBC, we considered 53 mouse Ccn6$^{fl/fl}$ and 120 human tumor upregulated significant proteins and 118 mouse and 353 human tumor downregulated significant proteins that were differentially expressed relative to their respective normal tissues. The cutoff for significant proteins was $p$-value < 0.05, $q$-value < 0.1, fold change (log2-FC) >1, which generated the 17-protein upregulated and 19-protein downregulated signatures.

**Reporting summary**. Further information on research design is available in the Nature Research Reporting Summary linked to this article.

## Data availability

All mass spectrometry proteomics data including human and mouse data sets were deposited to the ProteomeXchange Consortium through the PRIDE partner repository with identifier PXD014414. As the informed consent obtained from MBC patients does not allow for public deposition of the sequencing data, the WES sequencing data can be communicated upon reasonable request to C.G.K. All other data are available from the corresponding authors on reasonable request.

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

## Acknowledgements

We thank members of our laboratories for their critical reading of this paper and useful suggestions. We thank Drs. Sofia Merajver, Jennifer Linderman, and Shuichi Takayama for helpful discussions during the execution of the project. This research was supported by the National Institutes of Health National Cancer Institute R01 CA107469 (C.G.K.), R01 CA125577 (C.G.K.), U24 CA210967 (A.N.), R01 GM094231 (A.N.), T32 CA140044 (S.D.), and the University of Michigan Rogel Cancer Center Support Grant (P30 CA46592).

## Author contributions

S.D., M.E., and C.K. conceived the study; S.D., M.E., C.K., and A.N. designed experiments, performed the analyses, and interpreted the data. P.A., M.W., and A.C.-M. provided a subset of human tissue samples. C.K. performed histopathological evaluation of all tissues and selected areas for further analyses. B.B. performed sample preparation, V.B. performed data acquisition, F.L. and H.C. developed proteomics analysis tools, and S.T. prepared DNA samples. S.D. performed data processing, quantitative analysis, and interpretation of results, wrote the paper and prepared the figures. C.K., M.E., and A.N. assisted with the figures, and wrote the paper with input from all authors.

## Competing interests

The authors declare no competing interests.
