## [Peer Review File · Nature Communications]

Reviewers' comments:

Reviewer #1 (Remarks to the Author):

Djomehri et al present a proteomic study of clinical metaplastic breast carcinoma (MBC) samples, an aggressive rare form of triple negative breast cancer (TNBC), that they compare to other more frequently occurring TNBCs. The study comprises a limited set of tumor and normal samples collected from 20 patients, including 14 MBC cases of spindle (n=6), squamous (n=4) and sarcomatoid (n=4) histology, and 6 non-metaplastic TNBC or 6 normal adjacent tissues for comparisons.

For proteome profiling, tandem mass tags (TMT) were used for relative quantification at a moderate coverage of 5,798 proteins across all samples. It is important to note that imputation was used for most of the presented analyses, to overcome the limitations of missing values in these datasets. The degree to what extent missing values were imputed and how that affects the main conclusions of the study are not well described.

Another limitation of the study is that only proteomic data for all tumor samples are presented, but no genomic data. It is not known if the tumors chosen for this study represent common cases for the three histological MBC subtypes with respect to mutation status of the most frequently mutated breast cancer genes, such as PIK3CA, TP53, or CDKN2A, or if the selected tumors show mutation-specific effects that may be a confounding factor for the presented MBC subtype-specific effects.

The MBC-specific protein signatures are in general of interest, in particular the epithelial-to-mesenchymal transition (EMT) pathway effects and the MBC-specific signature of 15 genes that overlaps to a RNA-based MBC signature of a separate study and cohort. A smaller group of 6 out of these 15 genes overlaps also to a signature of a MBC mouse model (MMTV-cre;Ccn6fl/fl). A limitation hereby is, that the comparison of proteomic to RNA MBC signatures was done on different cohorts/sample sets, therefore it is unclear if the minimal overlap is driven by differences between protein and RNA regulation or differences between the cohorts. The cohort effects are more likely to make a bigger difference here. The comparison to the mouse model is also confounded, since no proteomics data were generated for the mouse model.

In summary, this study shows some interesting proteomic signatures for a rare, but very aggressive subtype of breast cancer. Due to limitations in sample size, proteomics coverage, and lack of genomics data, this manuscript is more suited for a specialized cancer research journal.

Major Points of concern:

Page 2, abstract: “Proteogenomics identified a 15 protein MBC subtype-specific signature”

Usually, the term “proteogenomics” is used when proteomic and genomic datasets for the same samples are compared. Here the comparison is across studies to different patient cohorts and RNA data. A major weakness of the study is that no genomics datasets were presented for the MBC.

Figure 7. “Proteogenomic associations between human and mouse MBC identify new subtype-specific biomarkers and therapeutic targets. a. 32 common genes/protein were found between the human MBC proteome (this study; 14 MBC tumors) and the human MBC transcriptome (microarray dataset with 9 MBC tumors; Hennessy 2009). We overlapped 960 MBC differentially expressed proteins and 827 MBC differentially expressed genes.”

The overlap between the RNA- and protein-based signature is minimal here, probably due to the different cohorts. What are the differences between the cohorts with respect to tumor stage, metastasis status, and mutation status of breast cancer-relevant genes?

Page 6: “We performed data imputation using

multivariate imputation by chained equations in R, followed by batch correction in R,

and the resulting batch-corrected, data-imputed expression matrix was used for all

downstream analyses (Fig. 2).”

The authors should state what percentage of values for the entire dataset were imputed. In particular, for all main MBC signatures it is important to show heatmaps without imputed values to make a visual inspection of results possible. For example, do Figures 6 and 7 include imputed values? If so, make them visible in the heatmaps or leave them out.

Page 4: A new proteomics data analysis pipeline “Philosopher/TMT-Integrator” was used for the data processing and analysis. Is this the first report of this pipeline? If so, it would be useful if the authors could benchmark their new tool with respect to identification numbers and quantification accuracy to already existing tools, such as the Max Quant software package.

Page 8: “Overall, 85% of the proteins identified were downregulated and 15% upregulated compared to normal breast.”

Samples were normalized for total protein content per sample, correct? The authors should provide an explanation why 85% of proteins were downregulated in MBC samples. Looking at Fig. 1a, it would be surprising if TNBC samples showed a downregulated proteome as compared to normal samples, since the cellularity is higher. Are the downregulation effects observed for MBC related to

the lower neoplastic cellularity in those specimens? Did the authors look at a correlation to neoplastic cellularity as defined by histopathological review?

Minor points of concern:

Page 2: "MBC and TNBC share downregulated angiogenesis, stemness and metabolic processes"

These effects were observed in comparison to normal tissue? Downregulated metabolic processes sound more interesting than the obvious effect of surrounding normal mammary fat tissue containing more lipid metabolism components. What metabolic processes are described here?

Page 7: "Differential expression analyses

show that metastatic MBC has upregulation of 69 and 113 proteins compared to normal breast and non-metastatic MBC, respectively, which comprise nuclear metabolism, cell cycle, DNA conformation changes, RNA processing, and chromatin organization."

This sentence is confusing, are pathways that are up in MBC compared to normal or TNBC the same? In general, the comparison of metastatic vs non-metastatic MBC should be more informative than the comparison of metastatic MBC to normal tissue.

Page 32: Figure 1B shows 15 MBC instead of 14 clinical samples as described in the text on page 5. Why were samples excluded?

Figure 2: Shows mostly quality control analyses and could be moved to supplements.

Reviewer #2 (Remarks to the Author):

Djomehri, et al present a study summarizing the proteomic landscape of metaplastic breast carcinomas. This is an under-studied subpopulation of breast cancers with poor outcome, which highlights that this study has the potential to be of high interest. Additionally, the proteomics dataset is novel and was generated with a robust, thoughtful experimental design. Several weaknesses in the data analysis and presentation need to be addressed prior to publication. These are described in detail below.

Major points

MBC is subtype of TNBC, so the rationale for many comparisons against normal breast tissue is unclear. Much more compelling are comparisons of MBC vs. TNBC as well as comparisons across MBC subtypes – such comparisons zero-in on key biological features that differ across the related subtypes. Further, the continual shift between reference frames is confusing and not always clearly stated (e.g., Discussion p.12, paragraph 2: “...MBC subtypes exhibit specific upregulated protein pathways.”) I suggest moving all comparisons against normal tissue to the supplement.

A major claim the authors would like to make is that the findings of differential proteomic activity across MBC subtypes is clinically relevant, however there are no analyses to strongly support this. The prognosis survival curves begin to get at this idea, however it would be much more compelling if the authors could suggest therapeutically-relevant signaling pathways operable in each of the subtypes. Such information could be functionally tested in other studies and if the findings validate, could ultimately be used to inform clinical decision making. Without any attempt to get to such an association, the authors should narrow their claims.

Related to the above point, I don't find the GO functional process enrichment analyses to be particularly compelling, and the authors provide very little interpretation of these results.

The analysis of TCGA MBC transcriptional data (Fig 7) is a key aspect of the study, as the authors use this to identify biomarkers associated with outcome. I have several concerns with this analysis and how it is presented. First, it is unclear how the authors mapped genes to proteins and how the differential analyses were performed. Second, the survival analysis curves show some that a few biomarkers can be used to stratify patients based on outcome, however the authors need to confirm that there are no differences in outcome based on subtype alone. If so, then the biomarkers may be simply associated with subtype, rather than outcome per se.

Based on the authors prior work with CCN6 pathway genes, I expected that they would assess whether these key genes also have prognostic implications.

Minor points

Figure 6 and 7: Heatmaps that include values within the cells are a bit cumbersome. I suggest presenting data with more traditional heatmaps without the values indicated. Additionally, vertical and horizontal bars that separate gene groups and sample groups would further aid in interpretation.

Methods for analysis are very sparse and the exact approach for the various analyses presented in the figures are not always clear.

Results section p.6, first full paragraph ("Upon inspection...") is principally methods details that could be moved to Methods section.

Reviewer #3 (Remarks to the Author):

The manuscript "Quantitative proteomic landscape of metaplastic breast carcinoma..." by Djomehri et al has been reviewed for publication in Nature Communications. The report describes novel proteomics data on a panel of breast cancers of the rare but highly aggressive metaplastic histology subtype, using advanced proteomic approaches that quantify more than 5000 proteins. Comparisons to normal breast tissue and general cases of TBNC are made. A key finding of this study is that three sub-morphologies of metaplastic breast cancer, spindle, squamous and sarcomatoid histologies, show distinct proteomic profiles. Furthermore, the data provides new clues to diagnostic and targetable pathways for these highly aggressive breast cancers. Overall, the report provides new data that represents an advance in understanding likely to influence thinking in the field of metaplastic breast cancer. However, several concerns need to be addressed.

1) The report is based on a very limited set of 14 MBC cases, which are further subdivided into three small groups of 6 spindle, 4 squamous and 4 sarcomatoid cases. There is a concern about overfitting of a large number of protein data across these small subgroups of cases. The authors should provide false discovery rates in tables that present univariate p-values to the extent possible.

2) Along the same lines, the further subgroup analysis of 4 MBC cases that had distant metastases, to identify candidate metastasis-associated proteins, seems both underwhelming (Supp fig 4) and

greatly underpowered and should be removed unless appropriate statistics support to importance of the results of this subanalysis (page 7-8, Supp fig 4 and supp table 3).

3) Survival analyses using Kaplan-Meier plots for Fig 7e and Table 3 - There is no description of how cutpoints for high and low marker expression is determined. Is this just cut at median (50% high and 50% low)? Consider using an optimal cutpoint selection procedure for these continuous variables, ideally with a test and validation set. Also please provide hazard ratios along with p-values for more meaningful interpretation.

4) Discussion is too lengthy and includes a lot of repetition of results and would greatly benefit from being condensed to half its size. However, a statement about limitations of this study should be included, including the analysis of limited cases of this rare MBC subtype and imputation of missing protein data.

Point by point response to the reviewers' comments. Please note that we state the reviewers' critiques exactly as they were stated in the letter.

REVIEWER 1

1. For proteome profiling, tandem mass tags (TMT) were used for relative quantification at a moderate coverage of 5,798 proteins across all samples. It is important to note that imputation was used for most of the presented analyses, to overcome the limitations of missing values in these datasets. The degree to what extent missing values were imputed and how that affects the main conclusions of the study are not well described. And

Page 6: "We performed data imputation using multivariate imputation by chained equations in R, followed by batch correction in R, and the resulting batch-corrected, data-imputed expression matrix was used for all downstream analyses (Fig. 2)." The authors should state what percentage of values for the entire dataset were imputed. In particular, for all main MBC signatures it is important to show heatmaps without imputed values to make a visual inspection of results possible. For example, do Figures 6 and 7 include imputed values? If so, make them visible in the heatmaps or leave them out.

Response: We appreciate the reviewer's comment. In the revised manuscript, we have specified the percentages of imputed values in each experiment. The new paragraph on Page 6 states "Upon inspection, experiments 1, 2, and 3 contained 18%, 14%, and 12% missing values, respectively (**Suppl Fig S1**). We performed data imputation using multivariate imputation..." The literature reports anywhere from 5 to 45% missing values are typically observed (Schmitt, Mandel and Guedj 2015).

As suggested, we also removed the previous heatmaps showing patient stratification signatures which did not show actual imputed expression values, but rather log₂-fold change values after imputation and differential expression. Of note, the human proteome heatmap (**Figure 2**) shows original raw data prior to imputation. In our revised manuscript, we confirmed in **Suppl Fig S1** that the distribution of samples with and without imputation led to unbiased results from batch correction, and show by statistical analysis how well the data imputed matrices correlate to experimental data. From significance tests, the performance of our parameter estimates was $p < 1.6E-03$ (**Suppl Fig S1B table**). We included additional analysis from the data imputation in **Suppl Fig S1** and provide further explanation about its usage in the methods section.

2. Another limitation of the study is that only proteomic data for all tumor samples are presented, but no genomic data. It is not known if the tumors chosen for this study represent common cases for the three histological MBC subtypes with respect to mutation status of the most frequently mutated breast cancer genes, such as PIK3CA, TP53, or CDKN2A, or if the selected tumors show mutation-specific effects that may be a confounding factor for the presented MBC subtype-specific effects.

And

Page 2, abstract: "Proteogenomics identified a 15 protein MBC subtype-specific signature" Usually, the term "proteogenomics" is used when proteomic and genomic datasets for the same samples are compared. Here the comparison is across studies to different patient cohorts and RNA data. A major weakness of the study is that no genomics datasets were presented for the MBC.

Response: Following the reviewer's suggestion, we have now performed a genomics analysis on our MBC patient cohort (**new Figure 6**). At present, mutation specific effects are not well described in MBC, with very few studies available (Cimino-Matthews 2017, Reis-Filho 2017). We carried out whole exome sequencing and analyzed mutational signatures present in individual MBC patients. Our new studies identified common MBC mutations that matched literature (e.g. TP53 and PI3K/MTOR) as well as notable MBC subtype-specific genomic alterations and novel mutations. Our **new Figure 6**

shows a genetic basis that clearly distinguishes spindle and squamous from sarcomatoid MBC, a novel finding, and reveal a therapeutically relevant set of targets (e.g. protocadherin (PCDH) gene mutations sarcomatoid MBC), which was previously unknown.

3. The MBC-specific protein signatures are in general of interest, in particular the epithelial-to-mesenchymal transition (EMT) pathway effects and the MBC-specific signature of 15 genes that overlaps to a RNA-based MBC signature of a separate study and cohort. A smaller group of 6 out of these 15 genes overlaps also to a signature of a MBC mouse model (MMTV-cre;Ccn6^{fl/fl}). A limitation hereby is, that the comparison of proteomic to RNA MBC signatures was done on different cohorts/sample sets, therefore it is unclear if the minimal overlap is driven by differences between protein and RNA regulation or differences between the cohorts. The cohort effects are more likely to make a bigger difference here. The comparison to the mouse model is also confounded, since no proteomics data were generated for the mouse model.

And

Figure 7. “Proteogenomic associations between human and mouse MBC identify new subtype-specific biomarkers and therapeutic targets. a. 32 common genes/protein were found between the human MBC proteome (this study; 14 MBC tumors) and the human MBC transcriptome (microarray dataset with 9 MBC tumors; Hennessy 2009). We overlapped 960 MBC differentially expressed proteins and 827 MBC differentially expressed genes.”

The overlap between the RNA- and protein-based signature is minimal here, probably due to the different cohorts. What are the differences between the cohorts with respect to tumor stage, metastasis status, and mutation status of breast cancer-relevant genes?

Response: We understand the reviewer and are grateful for the suggestions. In the revised manuscript, we have removed the comparison to RNA data from different patient cohorts which confounded the data, and added a new quantitative proteomics analysis of MMTV-cre;Ccn6^{fl/fl} spindle MBC tumors, which were developed in our lab (Martin EE 2017), **new Figure 7**. We overlapped the proteomics landscape of human and mouse spindle MBC tumors, and identified common deregulated protein pathways including epithelial to mesenchymal transition (EMT), Myc and E2F pathways, as well as a set of 36 significantly deregulated proteins (17 up regulated and largely ribosomal and translational proteins, and 19 down regulated and mainly involved in cell metabolism). These new data validate the utility of the mouse model in identifying markers for spindle MBC and nominate specific proteins with potential utility as biomarkers and therapeutic targets for spindle MBC.

We recognize that a limitation of our analysis is that the differential expression analysis of mouse proteomics was relative to normal tissues. However, there are no available models of other MBC subtypes at present.

4. Page 6: “We performed data imputation using multivariate imputation by chained equations in R, followed by batch correction in R, and the resulting batch-corrected, data-imputed expression matrix was used for all downstream analyses (Fig. 2).”

The authors should state what percentage of values for the entire dataset were imputed. In particular, for all main MBC signatures it is important to show heatmaps without imputed values to make a visual inspection of results possible. For example, do Figures 6 and 7 include imputed values? If so, make them visible in the heatmaps or leave them out.

Response: please see response #1.

5. Page 4: A new proteomics data analysis pipeline “Philosopher/TMT-Integrator” was used for the data processing and analysis. Is this the first report of this pipeline? If so, it would be useful if the authors could benchmark their new tool with respect to identification numbers and quantification accuracy to already existing tools, such as the Max Quant software package.

Response: The MSFragger, Philosopher and TMT-Integrator tools developed by the Nesvizhskii group can be found at: <https://msfragger.nesvilab.org>, <https://philosopher.nesvilab.org>, and <https://github.com/Nesvilab/TMT-Integrator>. Recently, these tools were described in a large-scale project published in Cell 2019 179(4):964-983.e31 (<https://doi.org/10.1016/j.cell.2019.10.007>).

6. Page 8: “Overall, 85% of the proteins identified were downregulated and 15% upregulated compared to normal breast.” Samples were normalized for total protein content per sample, correct? The authors should provide an explanation why 85% of proteins were downregulated in MBC samples. Looking at Fig. 1a, it would be surprising if TNBC samples showed a downregulated proteome as compared to normal samples, since the cellularity is higher. Are the downregulation effects observed for MBC related to the lower neoplastic cellularity in those specimens? Did the authors look at a correlation to neoplastic cellularity as defined by histopathological review?

Response: We understand the concern, and have evaluated the cellularity of the samples by expert histopathological review. After careful consideration we concluded that our initial observation of a majority of downregulated genes in cancer compared to normal tissues was biased as we had used a cutoff on the data. Thus, we have removed this observation from the revised manuscript. Instead, as the reviewer suggested, in the revised paper we performed multiple analysis comparing MBC with TNBC, and within MBC subtypes, using a statistical cutoff only for GO overrepresentation tests and without any cutoffs for the GSEA analysis.

7. Page 2: “MBC and TNBC share downregulated angiogenesis, stemness and metabolic processes.” These effects were observed in comparison to normal tissue? Downregulated metabolic processes sound more interesting than the obvious effect of surrounding normal mammary fat tissue containing more lipid metabolism components. What metabolic processes are described here?

Response: In our revised paper we have carried out a comprehensive analysis on the effects between tumors (as suggested by reviewer 2, point 1), and have included comparisons to normal tissues in the supplement (**Suppl Fig. S3**). Between tumors, we compared the proteomes of MBC with TNBC, and within MBC subtypes (**Figures 3-5**). We also compared each MBC subtype against TNBC (**Suppl Fig. S7**). The metabolic processes described are enriched pathways/significant terms from either gene ontology (GO) or gene set enrichment analysis (GSEA) including molecular signatures from hallmark, curated (canonical pathways including KEGG and REACTOME). For example, we found that compared to TNBC, MBC has significantly reduced metabolism (fatty acid metabolism, xenobiotic metabolism) and oxidative phosphorylation, and increased epithelial-mesenchymal transition.

8. Page 7: “Differential expression analyses show that metastatic MBC has upregulation of 69 and 113 proteins compared to normal breast and non-metastatic MBC, respectively, which comprise nuclear metabolism, cell cycle, DNA conformation changes, RNA processing, and chromatin organization.” This sentence is confusing, are pathways that are up in MBC compared to normal or TNBC the same?

In general, the comparison of metastatic vs non-metastatic MBC should be more informative than the comparison of metastatic MBC to normal tissue.

Response: The metastatic sample analysis has been removed at this time, considering the small size of metastatic patients (N=4), and by suggestion from Reviewer 3 (please see point 2 from reviewer 3).

9. Page 32: Figure 1B shows 15 MBC instead of 14 clinical samples as described in the text on page 5. Why were samples excluded?

Response: This has been corrected. The original patient cohort including 15 MBC is shown in **Table 1** (excluded sample is Case #5) and the human proteome heatmap (**Figure 2**) shows all 15 MBC with the excluded sample color coded in the legend. We excluded this sarcomatoid sample as both histology and proteomics confirmed that instead of tumor, we sampled a piece of normal tissue. This is described on page 6, “In addition, 1 of the 15 MBC samples were excluded from downstream analyses since it was confirmed by histology and proteomics that the piece cut for analysis was normal tissue.” Also, we described this in the Survey upon submission.

10. Figure 2: Shows mostly quality control analyses and could be moved to supplements.

Response: As suggested, we moved the quality control analyses for the human patient cohort to **Suppl Fig S1**. In addition, quality control analyses for the mouse model data set are included in Suppl Figure S8.

REVIEWER 2

1. Djomehri, et al present a study summarizing the proteomic landscape of metaplastic breast carcinomas. This is an under-studied subpopulation of breast cancers with poor outcome, which highlights that this study has the potential to be of high interest. Additionally, the proteomics dataset is novel and was generated with a robust, thoughtful experimental design. Several weaknesses in the data analysis and presentation need to be addressed prior to publication. These are described in detail below.

We thank the reviewer for the encouraging comment.

2. MBC is subtype of TNBC, so the rationale for many comparisons against normal breast tissue is unclear. Much more compelling are comparisons of MBC vs. TNBC as well as comparisons across MBC subtypes – such comparisons zero-in on key biological features that differ across the related subtypes. Further, the continual shift between reference frames is confusing and not always clearly stated (e.g., Discussion p.12, paragraph 2: “...MBC subtypes exhibit specific upregulated protein pathways.”) I suggest moving all comparisons against normal tissue to the supplement.

Response: We thank the reviewer for the suggestions. In the revised manuscript, we have performed a comprehensive analysis between MBC and TNBC, within MBC subtypes (**Figs 3-5, Suppl Figs S4-6**), and between MBC subtypes and TNBC (**Suppl Fig S7**). We also moved the comparisons to normal tissue to the supplement (**Suppl Fig S3**). The reference frames are now more consistent across these comparisons with unified headers (**Figs 3-5**).

3. A major claim the authors would like to make is that the findings of differential proteomic activity across MBC subtypes is clinically relevant, however there are no analyses to strongly support this. The prognosis survival curves begin to get at this idea, however it would be much more compelling if the

authors could suggest therapeutically-relevant signaling pathways operable in each of the subtypes. Such information could be functionally tested in other studies and if the findings validate, could ultimately be used to inform clinical decision making. Without any attempt to get to such an association, the authors should narrow their claims.

Response: Guided by the reviewer's critique, in our revised study we have carried out a comprehensive and in-depth analysis of therapeutically and/or functionally relevant signaling pathways from gene set enrichment analysis (GSEA) of hallmark pathways, canonical pathways (KEGG, REACTOME, etc) and gene ontology using the molecular signatures databases (MSigDB). We used the comparisons between tumors, delineated a common MBC vs. TNBC signature, and narrowed more distinct signatures within each MBC subtype. For example, compared to TNBC, MBCs have significantly upregulated EMT proteins and downregulated metabolic pathway proteins. Comparison within subtypes, revealed that spindle MBC are enriched for translation and ribosomal events, and elevated Myc and E2F targets, while squamous MBC show significant upregulation of apical junction proteins and inflammatory mediators, and PI3K signaling, and sarcomatoid MBC have elevated EMT and oxidative phosphorylation proteins.

4. Related to the above point, I don't find the GO functional process enrichment analyses to be particularly compelling, and the authors provide very little interpretation of these results.

Response: We agree and have revised the GO enrichment results to be complementary to the GSEA analysis. The GO analysis (from differential expression) showed an initial broad view of biological processes from both up- or downregulated profiles, followed by GSEA analysis (Hallmark, Curated, GO) to identify the activated and suppressed signaling pathways, including network topology graphs showing direct visualization of associated proteins in each pathway/process. We also agree that the GO enrichment tests are less compelling and it has been shown (Evangelou et al 2012, PLoS One. 2012;7(7):e41018) that it is less powerful than GSEA. Among competitive tests for pathway analysis, the Hypergeometric Test such as GO overrepresentation tests, are known to have the lowest power, whereas GSEA has higher power, is more robust, and does not require the use of arbitrary cutoffs placed on the data, which introduces biases in hypergeometric enrichment tests.

5. The analysis of TCGA MBC transcriptional data (Fig 7) is a key aspect of the study, as the authors use this to identify biomarkers associated with outcome. I have several concerns with this analysis and how it is presented. First, it is unclear how the authors mapped genes to proteins and how the differential analyses were performed. Second, the survival analysis curves show some that a few biomarkers can be used to stratify patients based on outcome, however the authors need to confirm that there are no differences in outcome based on subtype alone. If so, then the biomarkers may be simply associated with subtype, rather than outcome per se.

Response: We are grateful to the reviewer for pointing this out, and we agree. Thus, the analyses of TCGA / transcriptional data sets have been removed at this time. To identify novel biomarkers and potential targets of therapy for MBC subtypes, we performed in depth additional analyses on 1) comparison of the proteomics landscapes of MBC subtypes, 2) proteomics on a mouse model of spindle MBC, and 3) whole exome sequencing on our same MBC patient cohort. The proteomics analyses of human MBC revealed unique deregulated protein pathways specific to MBC subtypes (**new Figs. 3-5**). With the mouse proteome, we found new biomarkers that are therapeutically relevant for spindle MBC after overlapping them with the human spindle proteome (**new Fig. 7**). The whole exome sequencing studies in human MBC revealed mutational signatures within the MBC subtypes

(using the same patient’s healthy tissue as germline), and could correlate to an extent using the up- and downregulated signaling pathways from proteomics. Also, we related the mutational profiles to previous literature (**new Figure 6**). Therefore, our new studies uncover MBC subtype-specific gene mutations, and deregulated proteins and pathways that warrant further investigation as potential biomarkers or targets for precision medicine.

6. Based on the authors prior work with CCN6 pathway genes, I expected that they would assess whether these key genes also have prognostic implications.

Response: We appreciate the reviewer’s comment. We have analyzed the protein expression of CCN6 family genes as well as CCN6 downstream targets. Interestingly, we found that spindle MBC exhibit significant upregulation of HMGA proteins, especially HMGA2 (**Figure 3 “Spindle v Squamous” volcano plot**), which we recently found as one of the top upregulated genes in MMTV-cre;Ccn6^{fl/fl} and in human spindle MBC tumors (Martin EE, Oncogene 2017 and McMullen EM BCRT 2019). In

addition, we found that HMGA2 as well as another CCN6 target that we have identified, IGF2BP2/IMP2, are highest in human spindle MBCs compared to squamous and sarcomatoid MBC, based on log2 Fold changes of protein expression (see figure).

Figure. Patient stratified, unsupervised differential expression analysis showing that IGF2BP2 and HMGA2, two recently identified and novel CCN6-related MBC markers are significantly upregulated in MBC, especially in tumors with a spindle phenotype. One-way ANOVA statistical analyses were performed between subtypes, where p<0.05 was considered significant (*). These data are not included in the revised paper, as they need further investigation.

7. Figure 6 and 7: Heatmaps that include values within the cells are a bit cumbersome. I suggest presenting data with more traditional heatmaps without the values indicated. Additionally, vertical and horizontal bars that separate gene groups and sample groups would further aid in interpretation.

Response: We agree with the reviewer, and have edited the heatmaps.

8. Methods for analysis are very sparse and the exact approach for the various analyses presented in the figures are not always clear.

Response: We have revised the methods section and have edited the figures and legends so that they present the information clearly. In addition, we performed all the analyses in R and recommended packages from GO or GSEA documentation.

9. Results section p.6, first full paragraph (“Upon inspection...”) is principally methods details that could be moved to Methods section.

Response: We have edited this accordingly.

REVIEWER 3

The manuscript Quantitative proteomic landscape of metaplastic breast carcinoma...” by Djomehri et al has been reviewed for publication in Nature Communications. The report describes novel

proteomics data on a panel of breast cancers of the rare but highly aggressive metaplastic histology subtype, using advanced proteomic approaches to quantify more than 5000 proteins. Comparisons to normal breast tissue and general cases of TBNC are made. A key finding of this study is that three sub-morphologies of metaplastic breast cancer, spindle, squamous and sarcomatoid histologies, show distinct proteomic profiles. Furthermore, the data provides new clues to diagnostic and targetable pathways for these highly aggressive breast cancers. Overall, the report provides new data that represents an advance in understanding likely to influence thinking in the field of metaplastic breast cancer. However, several concerns need to be addressed.

We thank the reviewer for the positive comments on our study.

1. The report is based on a very limited set of 14 MBC cases, which are further subdivided into three small groups of 6 spindle, 4 squamous and 4 sarcomatoid cases. There is a concern about overfitting of a large number of protein data across these small subgroups of cases. The authors should provide false discovery rates in tables that present univariate p-values to the extent possible.

Response: We understand the concern. For protein inference, protein groups were initially filtered to 1% FDR using the target-decoy strategy and the best peptide approach as documented by Nesvizhskii et al 2003 and Shanmugam et al 2014. For each TMT experiment, peptides assigned to unique or razor peptides were additionally filtered to 1% PSM-level FDR for extracting only high confidence proteins. The issue of overfitting is present but is alleviated by having a well-defined proteomic study design, powered by statistically sound software tools.

2. Along the same lines, the further subgroup analysis of 4 MBC cases that had distant metastases, to identify candidate metastasis-associated proteins, seems both underwhelming (Supp fig 4) and greatly underpowered and should be removed unless appropriate statistics support the importance of the results of this subanalysis (page 7-8, Supp fig 4 and supp table 3).

Response: Following the suggestion, we have removed these analyses from the revised manuscript.

3. Survival analyses using Kaplan-Meier plots for Fig 7e and Table 3 - There is no description of how cutpoints for high and low marker expression is determined. Is this just cut at median (50% high and 50% low)? Consider using an optimal cutpoint selection procedure for these continuous variables, ideally with a test and validation set. Also please provide hazard ratios along with p-values for more meaningful interpretation.

Response: We appreciate the comment, and have removed the analysis of published transcriptional data sets (also please see reviewer 1, point 2). In this revised manuscript and following the suggestions, we have focused on elucidating the mutational landscape of MBC using whole exome sequencing, and on performing detailed quantitative proteomics, which led to the discovery of gene mutations and deregulated protein pathways between MBC and TNBC, and within MBC subtypes, which warrant further detailed studies to determine their biomarker and therapeutic potential.

4. Discussion is too lengthy and includes a lot of repetition of results and would greatly benefit from being condensed to half its size. However, a statement about limitations of this study should be included, including the analysis of limited cases of this rare MBC subtype and imputation of missing protein data.

Response: We appreciate the suggestion and have edited the discussion accordingly. As requested, we have added a statement about limitations as follows. “Our study has several limitations including small cohort size, and pathologically complex human samples. However here we have captured in a single workflow a very well-defined proteomics experimental design powered by statistically sound analyses using Philosopher and TMTIntegrator, which allow for systematic analysis of subtype-specific signaling targets.”

References cited in this response to reviewers

- Cimino-Mathews, A. et al. A clinicopathologic analysis of 45 patients with metaplastic breast carcinoma. *Am. J. Clin. Pathol.* 145, 365–372 (2016).
- Ng, C. et al. The landscape of somatic genetic alterations in metaplastic breast carcinomas. *Clin. Cancer Res.* 23, 3859–3870 (2017)
- Martin, E.E. et al. MMTV-cre;Ccn6 knockout mice develop tumors recapitulating human metaplastic breast carcinomas. *Oncogene* 36, 2275–2285 (2017).
- Schmitt P, Mandel J, Guedj M (2015) A Comparison of Six Methods for Missing Data Imputation. *J Biom Biostat* 6:224. doi: 10.4172/2155-6180.1000224.
- Clark D.J. et al. Integrated Proteogenomic Characterization of Clear Cell Renal Cell Carcinoma. *Cell.* 2019 Oct 31;179(4):964-983.e31. doi: 10.1016/j.cell.2019.10.007.
- Evangelou et al. Comparison of Methods for Competitive Tests of Pathway Analysis. *PLoS One.* 2012;7(7):e41018
- McMullen, E. R. et al. CCN6 regulates IGF2BP2 and HMGA2 signaling in metaplastic carcinomas of the breast. *Breast Cancer Res. Treat.* 172, 577–586 (2018).
- Nesvizhskii, A. I., Keller, A., Kolker, E. & Aebersold, R. A Statistical Model for Identifying Proteins by Tandem Mass Spectrometry abilities that proteins are present in a sample on the basis. *Anal. Chem.* 75, 4646–4658 (2003).
- Shanmugam, A. K., Yocum, A. K. & Nesvizhskii, A. I. Utility of RNA-seq and GPMDB protein observation frequency for improving the sensitivity of protein identification by tandem MS. *J. Proteome Res.* 13, 4113–4119 (2014).

REVIEWERS' COMMENTS:

Reviewer #1 (Remarks to the Author):

The authors have properly addressed all my points of concern and improved their manuscript by adding two new main figures. While the genomic analyses in Fig6 help to better understand the MBC cohort, the new Fig7 shows a better comparison of human and mouse model MBC tumors.

Reviewer #2 (Remarks to the Author):

Djomehri, et al have addressed all my concerns raised in the initial review. I believe the new manuscript is greatly strengthened. Below are a few minor stylistic comments that I think could be addressed to improve the readability of the paper and figures.

Methods details in figure legends are not needed if these are in methods section. Perhaps the editor can comment on journal style.

Axis labels missing on several figures: Fig 3 barplots; Fig 4 dotplots

Fig 2: Scale bar on heatmap missing

Fig 3: Right panel: It would be helpful to indicate the direction of enrichment for each gene set

Fig 4: It is not clear what the relationship is between barplot and scatterplot. Without labels, the GSEA barplot is not very informative, and so the authors should consider labelling it or removing it. In the right panel, to aid in interpretability, I suggest grouping the terms by whether they are up- or down-regulated.

Figure 6d: Legend refers to enrichment analysis, which I assume is presented in 6e. If so, it would be helpful to mention that, or else move all mention of enrichment analysis to 6e.

Reviewer #3 (Remarks to the Author):

The authors have adequately addressed my concern. The revised manuscript (including responses to other reviewers' suggestions), is greatly strengthened.

Point by point response to reviewers' comments.

(Note that the reviewer's comments are unedited)

Reviewer#1 (Remarks to the Author)

The authors have properly addressed all my points of concern and improved their manuscript by adding two new main figures. While the genomic analyses in Fig6 help to better understand the MBC cohort, the new Fig7 shows a better comparison of human and mouse model MBC tumors. **We thank the reviewer.**

Reviewer #2 (Remarks to the Author):

Djomehri, et al have addressed all my concerns raised in the initial review. I believe the new manuscript is greatly strengthened. Below are a few minor stylistic comments that I think could be addressed to improve the readability of the paper and figures.

1. Methods details in figure legends are not needed if these are in methods section. Perhaps the editor can comment on journal style. **We have edited this according to the editorial requests.**
2. Axis labels missing on several figures: Fig 3 barplots; Fig 4 dotplots. **We have added the missing labels.**
3. Fig 2: Scale bar on heatmap missing. **We have added the scale bar.**
4. Fig 3: Right panel: It would be helpful to indicate the direction of enrichment for each gene set. **The direction of enrichment is not applicable for barplots since we grouped both down- and upregulated proteins for this enrichment analysis.**
5. Fig 4: It is not clear what the relationship is between barplot and scatterplot. Without labels, the GSEA barplot is not very informative, and so the authors should consider labelling it or removing it. In the right panel, to aid in interpretability, I suggest grouping the terms by whether they are up- or down-regulated. **We agree with the reviewer. To make the grouping clearer, we have indicated this relationship between barplot and scatterplot with labeling upregulated and downregulated as UP (*) or DOWN (**).**
6. Figure 6d: Legend refers to enrichment analysis, which I assume is presented in 6e. If so, it would be helpful to mention that, or else move all mention of enrichment analysis to 6e. **We have edited the legend accordingly.**

Reviewer #3 (Remarks to the Author):

The authors have adequately addressed my concern. The revised manuscript (including responses to other reviewers' suggestions), is greatly strengthened. **We thank the reviewer.**